# Generation as Search Operator for Test-Time Scaling of Diffusion-Based Combinatorial Optimization

**Yang Li**[12†], **Lvda Chen**[1†], **Haonan Wang**[1], **Runzhong Wang**[1], **Junchi Yan**[12] *

[1]School of Computer Science & School of Artificial Intelligence, Shanghai Jiao Tong University
[2]Shanghai Innovation Institute
{yanglily,chenlvda,greedycow,runzhong.wang,yanjunchi}@sjtu.edu.cn
https://github.com/Thinklab-SJTU/GenSCO

## Abstract

While diffusion models have shown promise for combinatorial optimization (CO), their inference-time scaling cost-efficiency remains relatively underexplored. Existing methods improve solution quality by increasing denoising steps, but the performance often becomes saturated quickly. This paper proposes GenSCO to systematically scale diffusion solvers by an orthogonal dimension of inference-time computation beyond denoising step expansion, i.e., search-driven generation. GenSCO takes generation as a search operator rather than a complete solving process, where each operator cycle combines solution disruption (via local search operators) and diffusion sampling, enabling iterative exploration of the learned solution space. Rather than over-refining current solutions, this paradigm encourages the model to leave local optima and explore a broader area of the solution space, ensuring a more consistent scaling effect. The search loop is supported by a search-friendly solution-enhancement training procedure that incorporates a rectified flow model learning to establish diffusion trajectories between suboptimal solutions and the optimal ones. The flow model is empowered by a lightweight transformer architecture to learn neural ODEs that linearize solution trajectories, accelerating convergence of the scaling effect with efficiency. The resulting enhanced scaling efficiency and practical scalability lead to synergistic performance improvements. Extensive experiments show that GenSCO delivers performance improvements by orders of magnitude over previous state-of-the-art neural methods. Notably, GenSCO even achieves significant speedups compared to the state-of-the-art classic mathematical solver LKH3, delivering a $141\times$ speedup to reach 0.000% optimality gap on TSP-100, and approximately a $10\times$ speedup to reach 0.02% on TSP-500.

## 1  Introduction

Combinatorial Optimization (CO), which seeks optimal solutions in discrete spaces under complex constraints, underpins critical applications in vehicle routing [1, 2, 3], chip design [4, 5], and drug discovery [6]. However, the NP-hard nature of many CO problems renders them computationally intractable, traditionally requiring hand-crafted heuristics that demand substantial domain expertise and time to design. Recent advances in deep learning have revolutionized this landscape by automating heuristic design through data-driven neural solvers [7, 1, 8, 9, 10]. These ML-based approaches minimize human intervention while achieving competitive or superior solution quality and solving speed, particularly for instances following structured distributions. By learning generalizable frameworks directly from data, they also adapt seamlessly to unseen problem variants.

---

*Correspondence author. † denotes equal contribution. Runzhong Wang is currently affiliated with MIT; part of this work was done when he was at SJTU.

39th Conference on Neural Information Processing Systems (NeurIPS 2025).

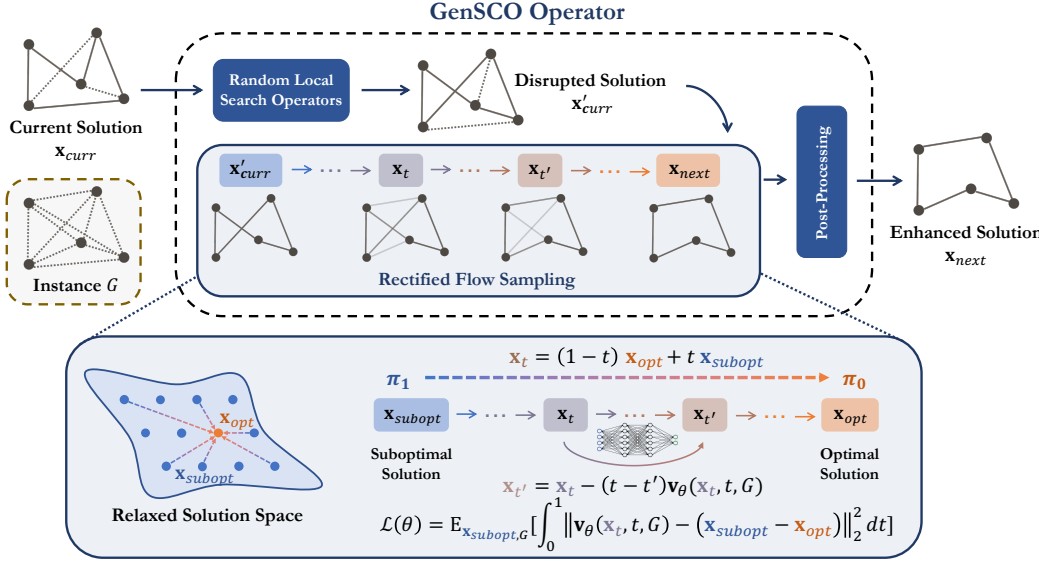

Figure 1: GenSCO treats diffusion sampling as a search step, where each step combines solution disruption through random local search operators and solution enhancement via rectified flow sampling. This GenSCO operator can be applied iteratively (or in parallel as well) to effectively explore the solution space with a stable scaling effect. The flow model learns neural ODEs that linearize the transition between suboptimal solutions and the optimal solution.

Learning-based CO solvers typically employ neural networks to generate solutions either by directly minimizing objective scores via reinforcement learning [1, 2, 11, 12, 13, 14, 15, 16] or by aligning predictions with reference solutions via supervised learning [17, 8, 18, 19, 20, 21, 22]. Recently, generative models, diffusion models in particular, have shown promise in solving CO with potent representational capabilities and informative distribution estimation, which models CO as a conditional generation task for learning solution distributions conditioned on given instances [21, 23, 9, 16]. Despite their promise, diffusion models primarily focus on enhancing the prediction accuracy of a single generation round, relying on the model's expressive power to refine solution feasibility and quality through multi-step denoising. While this framework inherently supports flexible inference-time computation via adjustable denoising steps, empirical evidence reveals a bottleneck in scaling up computation, i.e., scaling the number of steps yields rapidly diminishing returns, with performance saturating after only a few steps of computation. Given that computational efficiency is central to CO, where the goal is to achieve the optimal solution in the shortest possible time, this observed saturation effect represents a critical yet underexplored opportunity for substantial performance enhancement.

Recent works [9, 24] leverage diffusion models' distributional modeling capabilities to explore solution spaces during inference through alternating phases of disruption (perturbing candidate solutions) and reconstruction (guided by objective gradients). Our empirical results (as presented in Fig. 2) show that this target for solution space exploration provides superior scalability compared to simply increasing denoising steps, yet its full potential remains underexploited due to computationally expensive gradient updates during inference, creating a critical scalability bottleneck. In this work, we propose GenSCO that systematically redesigns both the diffusion architecture and the solving pipeline particularly for search purposes, aiming to maximize the inference-time scaling effects regarding computation. We propose to integrate diffusion-based generation into a novel search operator, where each operator performs three key functions in sequence as shown in Fig. 1: solution disruption with controllable local search operators to escape from local minima and introduce diversity, solution enhancement through a solution-to-solution generation, and post-processing to ensure feasibility. This operator can be applied iteratively (or in parallel as well) to explore the solution space.

For cost-effectiveness and performance scaling, our design adheres to two core principles, i.e., ensuring stable per-cycle improvement and minimizing per-cycle computational cost to enhance scalability. Rather than focusing on incremental improvements from extended denoising to emphasize over-parameterized single-round generation, we reallocate computation to higher-level refinement cycles. Each cycle ensures reliable solutions with minimal inference-time cost, while repeated

cycles effectively unlock high-quality regions of the solution space, collectively enhancing both efficiency and solution quality. To maximize per-cycle improvement without introducing additional computational cost during inference, GenSCO adopts a rectified flow embodiment of the diffusion model, which transitions from the prior noise-to-solution paradigm to a more search-friendly solution-to-solution refinement target. It formulates a bias-rectified training process where the model receives a worse-quality solution as the initial source noise and learns to denoise it to the optimal, which shifts the optimization burden to the training phase. This naturally introduces solution enhancement capability and enables adaptive control over the intermediate solution states during the search process. For scalability of search cycles, the rectified flow model leverages neural ordinary differential equations (ODEs) to learn nearly straight solution trajectories, minimizing truncation errors and enabling high-quality generations with fewer inference steps, which ensures reliable denoising while maintaining cost-efficiency. To further enhance scalability, we implement a lightweight transformer architecture that operates exclusively on node embeddings, which minimizes computational overhead while enabling thousands of evaluations within practical time budgets.

The highlights include: (1) We pioneer an inference-time scaling cost-efficiency perspective for diffusion solvers, challenging the conventional paradigm of allocating computation to maximize single-round generation quality. Instead, we propose a novel framework that treats generation as a search step, enabling systematic exploration of the solution space through refinement cycles. (2) We systematically redesign both the diffusion architecture and the solving pipeline specifically for search-driven optimization. Major designs include a solution-to-solution rectified flow model supported by a bias-rectified training objective, and a controllable search operator that enables efficient scaling. (3) Extensive experiments demonstrate that GenSCO significantly outperforms state-of-the-art methods in both solution quality and solving speed, achieving order-of-magnitude improvements.

## 2 Related Work

**Neural Combinatorial Optimization.** Learning-based methods for CO can be broadly categorized into constructive methods, improvement-based methods, and divide-and-conquer frameworks. Constructive methods generate solutions either autoregressively or non-autoregressively. Autoregressive methods [1, 2, 25, 11, 26, 22, 27] iteratively build solutions by selecting one variable at each step while maintaining feasibility. In contrast, non-autoregressive methods [8, 19, 28, 12, 21, 29, 30, 31, 32] directly predict the probabilities of variables belonging to the optimal solution in a single pass. These methods often require post-processing to enforce feasibility due to the potential infeasibility of raw predictions. Improvement-based solvers [33, 15, 34, 35, 36, 14] learn to iteratively refine a solution through local search operators guided by neural networks while preserving or restoring feasibility. Relatively orthogonal to pure solving methods, divide-and-conquer frameworks [35, 37, 38, 39, 40, 41] adopt a hierarchical strategy to address scalability challenges. By decomposing large-scale problems into smaller subproblems, these methods leverage existing solvers (either classical or neural) on the subproblems and aggregate the results into a global solution. Recent unified frameworks and benchmarks [42, 43, 44, 45] have been developed to systematically analyze the design space of CO approaches, yielding principled insights and practical guidelines for method design.

Generative modeling [46, 47, 48, 49, 50, 51, 52] has shown promise in CO solving [21, 9, 24, 16, 53, 54] and data supports [55, 56, 57]. The work [21] highlights the potential of generative CO, utilizing the powerful representational capabilities and informative distribution estimation of diffusion models to learn solution distributions. [16] promotes the diffusion framework to the unsupervised CO solving scenarios. Furthermore, [24] introduces a training consistency scheme ensuring that all noise trajectories for a certain graph converge to the same initial solution, thereby significantly reducing the number of denoising steps required. [9, 24] both introduce specific objective-guided gradient search during solving through re-generation within the noise-solution transition space to further explore the estimated solution distribution. While these techniques improve performance beyond denoising step scaling, they remain limited as auxiliary modules due to their reliance on costly gradient updates during inference, which creates a scalability bottleneck. Meanwhile, current models operate in the noise-to-solution transition space where the source noise distribution is stationary and intermediate states are untractable, lacking sufficient controllability.

**Diffusion Models and Flow Models.** Diffusion models have emerged as a powerful framework for generative modeling, involving a dual process: the forward diffusion process, which gradually adds noise to data, and the reverse denoising process, where a model learns to reconstruct the

original data from noisy versions. For Diffusion in continuous space [58, 59, 60, 61, 62, 63, 64], the solution trajectories can be modeled by Probability Flow ODE [65]. Similar paradigms have also been adopted for discrete data using binomial or multinomial/categorial noises [58, 66, 67]. Flow models [68, 69, 70, 71, 72] such as rectified flows [68] build upon the diffusion framework by focusing on learning transport maps between distributions as a flow that follows nearly straight paths defined via neural ODEs. Rectified flows can learn smooth ODE trajectories that are less susceptible to truncation error, which allows for high-quality samples with fewer inference steps than diffusion models. Moreover, they can be generalized to map two arbitrary distributions to one another, making them particularly suited for search-friendly solution-to-solution generation in CO.

# 3  Preliminaries and Problem Definition

**Combinatorial Optimization.** Following the conventions of [73, 74], we formalize CO over a family of problem instances represented as graphs $\mathcal{G}$, where each instance $G(V, E) \in \mathcal{G}$ consists of $V$ and $E$ denote the vertex set and edge set respectively. CO problems can be broadly classified into two types based on the solution composition: edge-decision and node-decision problems. Let $\mathbf{x} \in \{0, 1\}^N$ denote the optimization variable, where each entry with $1$ indicates that it is included in $\mathbf{x}$ and $0$ indicates the opposite. For edge-decision problems, $N = |V|^2$ and $\mathbf{x}_{i,j}$ indicates whether $E_{i,j}$ is included in $\mathbf{x}$. For node-decision problems, $N = |V|$ and $\mathbf{x}_i$ indicates whether $V_i$ is included in $\mathbf{x}$. The feasible set $\Omega$ contains all solutions $\mathbf{x}$ satisfying specific constraints. A CO problem on $G$ aims to find a feasible $\mathbf{x}$ that minimize the given objective function $l(\cdot; G) : \{0, 1\}^N \to \mathbb{R}_{\geq 0}$:

$$\min_{\mathbf{x} \in \{0,1\}^N} l(\mathbf{x}; G) \quad \text{s.t.} \quad \mathbf{x} \in \Omega \tag{1}$$

The Travelling Salesman Problem (TSP) is defined on complete undirected graphs, where vertices represent cities and edges have non-negative weights (e.g., distances). The goal is to find a minimum-weight Hamiltonian cycle. For Maximal Independent Set (MIS), given an undirected graph $G = (V, E)$, an independent set $S \subseteq V$ contains no adjacent vertices. MIS seeks the largest such set in $G$. The Maximum Clique (MCl) problem, given an undirected graph $G = (V, E)$, seeks a largest subset $C \subseteq V$ such that the subgraph induced by $C$ is complete.

**Diffusion Modeling for CO.** The goal of diffusion modeling for CO is to learn the distribution $p_\theta(\mathbf{x}_0|G)$ of high-quality solutions of instance $G$. This framework operates through a forward noising process that gradually corrupts an initial solution $\mathbf{x}_0 \sim q(\mathbf{x}_0|G)$ over $T$ timesteps to produce latent variables $\mathbf{x}_{1:T}$, and a learned reverse process that reconstructs the solution by parameterizing the joint distribution $p_\theta(\mathbf{x}_{0:T}) = \Pi_{t=1}^T p_\theta(\mathbf{x}_{t-1}|\mathbf{x}_t)$. The training optimization aims to align $p_\theta(\mathbf{x}_0|G)$ with the data distribution $q(\mathbf{x}_0|G)$ using the variational upper bound of the negative log-likelihood.

Current implementations [21, 9] employ either discrete or continuous diffusion approaches. In discrete diffusion, solutions are represented as $N$ one-hot vectors $\mathbf{x} \in \{0, 1\}^{N \times 2}$, with the forward process applying multinomial noise through $q(\mathbf{x}_t|\mathbf{x}_{t-1}) = \text{Cat}(\mathbf{x}_t; \mathbf{p} = \mathbf{x}_{t-1}\mathbf{Q}_t)$, where $\mathbf{Q}_t = \begin{bmatrix} \beta_t & 1 - \beta_t \\ 1 - \beta_t & \beta_t \end{bmatrix}$. This process gradually randomizes the variables according to the corruption rate $\beta_t$, eventually converging to a uniform stationary distribution. Continuous diffusion models, on the other hand, map discrete solutions to the continuous domain $[-1, 1]^N$ to enable the use of Gaussian noise. The forward process in this case follows $q(\mathbf{x}_t|\mathbf{x}_{t-1}) = \mathcal{N}(\mathbf{x}_t; \sqrt{1 - \beta_t}\mathbf{x}_{t-1}, \beta_t\mathbf{I})$, with the noise scale controlled by $\beta_t$ and the stationary distribution being Gaussian.

# 4  The GenSCO Framework: Generation as a Search Operator

In diffusion solvers, scaling the number of denoising steps is a conventional strategy to enhance single-round generation quality. By discretizing the diffusion process into finer time intervals, the solver theoretically achieves more precise noise estimation. However, as the number of steps increases, the solver's capacity to navigate high-dimensional, constrained solution spaces becomes the primary bottleneck rather than temporal resolution. The inherent task complexity leads to imperfect noise estimation, even with finer temporal granularity, and minor prediction errors accumulate across steps. Consequently, merely increasing computational resources for finer discretization yields diminishing returns, prompting the need for alternative scaling paradigms.

This section introduces the GenSCO framework, which treats generation as a search operator within the near-optimal solution space and scales solving performance through corresponding search cycles. To address the bottleneck of scaling single-round prediction precision, GenSCO shifts the paradigm to systematic exploitation of the learned high-quality solution distribution. The subsequent subsections first present a lightweight rectified model that learns solution-to-solution refinement, supporting near-optimal search cycles with flexible control over exploration. Then, the specific search-based inference pipeline that integrates learned refinement policies is introduced to achieve more effective scaling of solving performance with respect to computation.

## 4.1 Learning Solution Enhancement with Lightweight Rectified Flow Model

Existing diffusion approaches for CO often anchor their generative processes to stationary noise distributions to enable standard sampling paradigms, exhibiting a disconnect between the diffusion trajectory and the structured nature of solution spaces, i.e., the generative process operates in the noise space rather than within the (relaxed) solution space. The heavy de novo generation process relying on Gaussian initialization restricts both the controllability of the intermediate states within generation and the exploitation of prior solution knowledge. For the search purpose, we redefine the diffusion paradigm as a guided transition process between suboptimal and optimal solutions, a formulation that better aligns with heuristic search dynamics. We replace conventional noise injection with problem-specific local search operators (e.g., randomized 2Opt moves for TSP or variable flipping for MIS and MCl) to simulate possible suboptimal solutions that the model may encounter during inference. This enables adaptive control over source distribution to balance exploration-exploitation tradeoffs and utilization of existing solutions as diffusion initializations. To improve cost-efficiency and scalability, we implement a rectified flow model that learns straight-line interpolation paths between perturbed solutions and optimal targets, achieving both model compactness and accelerated sampling, facilitating the scaling of search operator calls during inference.

GenSCO adopts a continuous-time probability flow ODEs [65, 68], where the transformation between solution distributions occurs along a smooth trajectory parameterized by $t \in [0, 1]$. Given a graph instance $G$, denote the target point distribution of the optimal solution as $\pi_0$, the distribution of suboptimal (feasible) solutions as $\pi_1$. In practice, for TSP, we perform 2Opt operations on the optimal solution $\frac{N}{4}$ to $\frac{3N}{4}$ times, and for MIS and MCl, we randomly flip between $\frac{1}{4}$ and $\frac{3}{4}$ of the variables assigned with 1 to 0 to get suboptimal solutions. To facilitate smooth transitions between these solutions, we relax the $\{0, 1\}$-valued solutions to the interval $[0, 1]$ and treat the variables in the intermediate process as real-valued. Given solutions $\mathbf{x}_{\text{opt}} \sim \pi_0, \mathbf{x}_{\text{subopt}} \sim \pi_1$, the rectified flow induced from $(\mathbf{x}_{\text{opt}}, \mathbf{x}_{\text{subopt}})$ is an ordinary differentiable model (ODE) on time $t \in [0, 1]$,

$$\mathrm{d}\mathbf{x}_t = \mathbf{v}(\mathbf{x}_t, t, G)\mathrm{d}t \tag{2}$$

which converts solution $\mathbf{x}_{\text{opt}}$ from $\pi_0$ to a $\mathbf{x}_{\text{subopt}}$ following $\pi_1$. We establish the solution transition as the linear trajectory to guarantee fast convergence:

$$\mathbf{x}_t = t\mathbf{x}_{\text{subopt}} + (1 - t)\mathbf{x}_{\text{opt}}, t \in [0, 1] \tag{3}$$

In this case, the drift force $\mathbf{v}$ is set to drive the flow to follow the direction $(\mathbf{x}_{\text{subopt}} - \mathbf{x}_{\text{opt}})$ of the linear path pointing from $\mathbf{x}_{\text{opt}}$ to $\mathbf{x}_{\text{subopt}}$ as much as possible, i.e., $\frac{\mathrm{d}\mathbf{x}_t}{\mathrm{d}t} = \mathbf{v}(\mathbf{x}_t, t, G) = \mathbf{x}_{\text{subopt}} - \mathbf{x}_{\text{opt}}$. Since $\mathbf{x}_{\text{opt}}$ is unknown for simulating this ODE flow, we parameterize the velocity field $\mathbf{v}$ with a neural network with the following loss function:

$$\mathcal{L}(\theta) = \mathbb{E}_{\mathbf{x}_{\text{subopt}}, G} \left[ \int_0^1 \|\mathbf{v}_\theta(\mathbf{x}_t, t, G) - (\mathbf{x}_{\text{subopt}} - \mathbf{x}_{\text{opt}})\|_2^2 \, \mathrm{d}t \right]. \tag{4}$$

In implementation, we represent $\mathbf{v}_\theta(\mathbf{x}_t, t, G) = \mathbf{x}_{\text{subopt}} - \mathbf{f}_\theta(\mathbf{x}_t, t, G)$ where $\mathbf{f}_\theta(\mathbf{x}_t, t, G)$ directly predicts the optimal solution given the graph instance and the current state. Given a sequence of time points $1 = \tau_1 > \tau_2 > \cdots > \tau_{N_\tau - 1} > \tau_{N_\tau} = 0$, the generation process defines a solution enhancement process where the ODE can be solved via numerical integration, e.g., Euler's method:

$$\mathbf{x}_{\tau_{n+1}} = \mathbf{x}_{\tau_n} - (\tau_n - \tau_{n+1})\mathbf{v}_\theta(\mathbf{x}_{\tau_n}, \tau_n, G), \ \forall t \in \{\tau_1, \tau_2, \cdots, \tau_{N_\tau - 1}\} \tag{5}$$

A large value of $N_\tau$ results in accurate but slow simulations, while a small value of $N_\tau$ leads to faster but less accurate simulations. Since the probability flows are ideally modeled as straight paths, a small number of steps can still yield plausible results. To improve scaling by redistributing single-cycle computations across more cycles, we set $N_\tau = 4$ in the majority of our empirical evaluations.

**Architecture.** Prior work predominantly relies on deep anisotropic graph neural networks with edge gating mechanisms [8, 21] to support reliable single-round heatmap prediction, particularly for edge-based decision tasks like TSP. It incurs significant computational overhead due to its quadratic complexity in handling edge features, leading to the compounding latency from iterative denoising, severely constraining the scalability of such approaches as practical search operators.

To ensure scalability, we propose a lightweight architecture centered on node-only feature processing via a transformer-based architecture, eliminating the need for explicit edge feature computation. For input processing, node features undergo linear projection into the model's embedding dimension. The self-attention mechanism is augmented to explicitly encode graph structure through a simple yet effective modification to the attention logits computation:

$$\text{Attention}(\mathbf{Q}, \mathbf{K}, \mathbf{V}) = \text{softmax}\left(\mathbf{Q}\mathbf{K}^\top / \sqrt{d_k} + \lambda \mathbf{A}\right) \mathbf{V} \tag{6}$$

where $\mathbf{A}$ represents the adjacency matrix and $\lambda$ is a scaling parameter that balances the relative importance of learned attention patterns versus explicit graph connectivity. By convention, $\mathbf{Q}, \mathbf{K}, \mathbf{V}$ denote the query, key, value matrices for the vertices, respectively. $d_k$ denote the dimension of $\mathbf{K}$.

Our model operates solely on node features $\mathbf{h} \in \mathbb{R}^{N \times f}$, where $N$ is the number of nodes and $f$ the feature dimension. The transformer layers capture global dependencies through self-attention, after which task-specific outputs are generated. For node-wise decisions (e.g., MIS and MCl), we apply linear layers to project node embeddings directly into probability vectors. For edge-wise decisions (e.g., TSP), we compute pairwise connection logits (unnormalized log-probabilities) via an inner product $\mathbf{h}\mathbf{h}^\top \in \mathbb{R}^{N \times N}$, yielding a heatmap without explicit edge-level operations. This design preserves expressiveness while maintaining computational tractability in iterative refinement scenarios, enabling scalable operator calls (hundreds to thousands) during search.

## 4.2 Scaling Solving Performance with Search-Driven Generation

Recent advances in diffusion models for image generation overcome quality plateaus by strategically leveraging sampling stochasticity during inference [75]. Studies demonstrate that certain noise patterns yield superior outputs [76, 77], prompting methods to devote computation to identifying these preferable noise configurations through multi-sample exploration. Translating this insight to CO, where generative models approximate high-performance solution distributions, we argue that systematically exploiting the solution space, i.e., shifting from a focus on single-round prediction accuracy to exploring the breadth of the solution space, can unlock more effective search strategies. CO problems exhibit two distinctive characteristics over conventional generative tasks. The first one lies in the existence of deterministic solution verifiers, i.e., the optimization objective itself. This enables rapid quality evaluation of candidate solutions and facilitates flexible control over search processes through immediate feedback. Second, unlike generative tasks requiring broad mode coverage, CO prioritizes incremental refinement toward optimality. This permits a focused search paradigm where minor perturbations to near-optimal solutions supersede full-space exploration.

---

**Algorithm 1** GenSCO Framework for Solving

**Input:** Flow model $\mathbf{v}_\theta(\cdot, \cdot, \cdot)$, graph problem instance $G$, number of cycles $N_c$, sequence of time points $1 = \tau_1 > \tau_2 > \cdots > \tau_{N_\tau} = 0$

Randomly initialize solution $\mathbf{x}$
**for** cycle $c = 1$ to $N_c$ **do**
    $\mathbf{x}_{\tau_1} \leftarrow \text{Disrupt}(\mathbf{x})$
    **for** time step $n = 1$ to $N_\tau$ **do**
        $\mathbf{d}_{\tau_n} = \mathbf{v}_\theta(\mathbf{x}_{\tau_n}, \tau_n, G)$
        $\mathbf{x}_{\tau_{n+1}} = \mathbf{x}_{\tau_n} - (\tau_n - \tau_{n+1})\mathbf{d}_{\tau_n}$
    **end for**
    $\mathbf{x} \leftarrow \text{PostProcess}(\mathbf{x}_{\tau_{N_\tau}})$
**end for**
**Output:** Solution $\mathbf{x}$

---

Based on these considerations, we propose a neural-based local search framework that iteratively refines solutions within near-optimal regions supported by search-driven generation. This method introduces an orthogonal dimension of inference-time computation beyond denoising step expansion, i.e., search cycles. This approach leverages the solution-to-solution enhancement mapping (as introduced in Sec. 4.1) and the diversity of the learned solution space to design a structured search process. While a naive implementation might iteratively apply the learned solution enhancement mapping to the current solution, such a strategy risks stagnation at local optima. To address this, we introduce a search operator that synergizes two components, i.e., solution disruption (via classic local search operators) that perturbs solutions to escape suboptimal regions, and diffusion sampling that

leverages the generative prior to navigate toward high-potential regions. Raw model outputs require decoding (and optional local search post-processing) to become valid solutions. For a valid search operator, we tightly integrate decoding and refinement steps directly into the search operator, ensuring feasible intermediate states while preserving real-time feedback from solution quality evaluations.

As detailed in Alg. 1, the solving process initiates with a randomly initialized solution vector $\mathbf{x}$, obtained through greedy decoding applied to randomized variable heatmaps. Subsequently, the GenSCO framework performs $N_c$ search cycles. In each cycle, $\mathbf{x}$ undergoes disruption: for TSP, this involves executing between $\frac{N}{4}$ and $\frac{3N}{4}$ 2Opt operations to the existing solution; for MIS, a random selection of 25% to 40% of the variables assigned a value of 1 are flipped to 0, resulting in a disrupted solution $\mathbf{x}_1$; for MCl, 25% to 40% of the variables are randomly selected and flipped. To enhance the solution, $\mathbf{x}_1$ is treated as a relaxed solution within the domain $[0, 1]^N$. Given a sequence of time points $1 = \tau_1 > \tau_2 > \cdots > \tau_{N_\tau - 1} > \tau_{N_\tau} = 0$, the enhancement process is modeled by an ODE, which is solved through numerical integration, such as Euler's method, as expressed in Eq. 5. The outcome is an enhanced solution $\mathbf{x}_0$ residing within the relaxed solution space $[0, 1]^N$. To derive a feasible solution, $\mathbf{x}_0$ is interpreted as a variable heatmap, where each element represents the confidence level of selecting the corresponding edge or node. Greedy decoding is then applied to sequentially select edges or nodes with the highest confidence, provided no conflicts arise. For TSP, the 2Opt heuristic [78] is optionally employed within post-processing. Upon completion of $N_c$ search cycles, GenSCO reports the best solution identified during the search process. Notably, the entire search procedure is amenable to parallel execution, leveraging complementary search effects. The pipeline of the proposed operator is presented in Fig. 1.

## 5 Experiments

In this section, we empirically compare our proposed GenSCO with other learning-based and classical solvers on TSP, MIS and MCl instances with various sizes and distributions. Source code will be made publicly available.

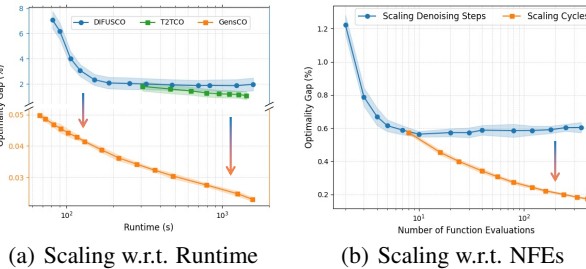

(a) Scaling w.r.t. Runtime (b) Scaling w.r.t. NFEs

### 5.1 Experiments for TSP

**Datasets.** A TSP instance includes $N$ 2-D coordinates and a reference solution. Instances are generated via uniformly sampling $N$ nodes from the unit

Figure 2: (a) Scaling curves for the traditional diffusion model, via denoising steps and gradient steps, and GenSCO through search cycles on TSP-1000. (b) The performance variation when scaling denoising steps in a single generation round and scaling search cycles in GenSCO.

square $[0, 1]^2$. The training sets consist of 1,280K instances for TSP-100, and 128K for TSP-500 and 1000. The test sets include 1,280 instances for TSP-100, and 128 instances for TSP-500 and 1000. Reference solutions are obtained by Concorde [79]. Results on TSPLIB[2] data are in the Appendix.

**Metrics.** 1) Objective: the average total distance or cost of the solved tours w.r.t. the corresponding instances; 2) Gap: the relative performance drop with respect to length compared to the global optimality or the reference solution; 3) Time: the computational time to solve the problems.

**Model Setting.** We use 4 denoising steps in a single round of flow generation and run 8 GenSCO operators in parallel. In the post-processing step, we apply greedy decoding with optional $N/50$ steps of 2Opt. The number of search cycles, denoted as $C$, is treated as a hyperparameter to control the extent of exploration. To configure the diffusion baselines, we adopt $T_s$ and $T_g$ to represent the number of inference steps in initial solution sampling and the number of gradient search steps [9, 24], respectively.

**Main Results for TSP.** Table 1 presents the solving performance comparison. As shown, GenSCO significantly outperforms all previous baselines and even LKH3 within its solving time. For example, on TSP-100, GenSCO attains an optimality gap of $0.000\%$ in just 6 seconds, whereas LKH3 [80] indeed requires 14.1 minutes to reach $0.000\%$ ($141\times$ slower), and the raw diffusion baseline DIFUSCO [21] takes 52.1 minutes to achieve 0.06%. Similarly, on TSP-500, GenSCO achieves 0.012% in 18 seconds, compared to DIFUSCO's 0.87% in 19.1 minutes, with a $72.5\times$ reduction in optimality gap and a $60.1\times$ speedup. Meanwhile, LKH3 requires 1.1 minutes ($3.7\times$ slower) to match this

---

[2]http://comopt.ifi.uni-heidelberg.de/software/TSPLIB95/

Table 1: Results on TSP. AS: Active Search, G: Greedy, S: Sampling Decoding, BS: Beam Search.

| Method | TSP-100 (1280 inst.) | | | TSP-500 (128 inst.) | | | TSP-1000 (128 inst.) | | |
|---|---|---|---|---|---|---|---|---|---|
| | Obj. | Gap | Time | Obj. | Gap | Time | Obj. | Gap | Time |
| *Mathematical Solvers or Heuristics* | | | | | | | | | |
| Concorde [79] | 7.76 | 0.000% | 5.1m | 16.55 | 0.000% | 39.8m | 23.12 | 0.000% | 3.0h |
| LKH-3 [80] (trials=32) | 7.76 | 0.009% | 52s | 16.55 | 0.024% | 42s | 23.13 | 0.035% | 2.2m |
| LKH-3 [80] (trials=64) | 7.76 | 0.006% | 59s | 16.55 | 0.015% | 50s | 23.12 | 0.025% | 2.5m |
| LKH-3 [80] (trials=128) | 7.76 | 0.003% | 1.2m | 16.55 | 0.011% | 1.1m | 23.12 | 0.015% | 3.1m |
| LKH-3 [80] (trials=256) | 7.76 | 0.002% | 1.6m | 16.55 | 0.006% | 1.6m | 23.12 | 0.009% | 4.1m |
| LKH-3 [80] (trials=512) | 7.76 | 0.001% | 2.5m | 16.55 | 0.002% | 2.5m | 23.12 | 0.005% | 6.2m |
| LKH-3 [80] (trials=4096) | 7.76 | 0.000% | 14.1m | 16.55 | 0.000% | 15.0m | 23.12 | 0.001% | 36.3m |
| *Non-Generative Neural Solvers* | | | | | | | | | |
| AM [1] BS | 7.95 | 2.48% | 12.9m | 19.53 | 18.03% | 2.8m | 29.90 | 29.24% | 12.6m |
| POMO [2] ×8 augment | 7.77 | 0.13% | 8s | 20.22 | 22.19% | 1.0m | 32.55 | 40.57% | 8.0m |
| POMO+EAS [81] AS+G | 7.76 | 0.05% | 38.4m | 24.54 | 48.22% | 11.6h | 49.56 | 114.36% | 63.5h |
| GCN [8] BS | 8.41 | 8.38% | 6.0m | 30.37 | 83.55% | 38.0m | 51.26 | 121.73% | 51.7m |
| DIMES [12] S+2Opt | 7.98 | 2.88% | 5.0m | 17.64 | 6.56% | 1.1m | 24.81 | 7.29% | 2.9m |
| DIMES [12] AS+S+2Opt | 7.93 | 2.18% | 3.2h | 17.29 | 4.48% | 2.1h | 24.32 | 5.17% | 4.5h |
| LEHD [82] PRC 100 | 7.76 | 0.01% | 1.8m | 16.61 | 0.34% | 8.0m | 23.44 | 1.22% | 43.0m |
| LEHD [82] PRC 1000 | 7.76 | 0.002% | 16.9m | 16.58 | 0.17% | 1.2h | 23.32 | 0.72% | 7.0h |
| BQ-NCO [22] BS | 7.76 | 0.01% | 4.1m | 16.64 | 0.55% | 15m | 23.47 | 1.38% | 38m |
| GLOP [39] | – | – | – | 16.91 | 1.99% | 1.5m | 23.84 | 3.11% | 3.0m |
| UDC [40] | – | – | – | 16.78 | 1.58% | 4.0m | 23.53 | 1.78% | 8.0m |
| COExpander [53] | 7.76 | 0.01% | 3.84m | 16.59 | 0.25% | 1.41m | 23.27 | 0.64% | 5.2m |
| *Generative Neural Solvers* | | | | | | | | | |
| DIFUSCO ($T_s$=50) [21] G+2Opt | 7.78 | 0.28% | 18.9m | 16.80 | 1.50% | 4.7m | 23.55 | 1.89% | 14.4m |
| DIFUSCO ($T_s$=100) [21] S+2Opt | 7.76 | 0.06% | 30m | 16.69 | 0.87% | 19.1m | 23.42 | 1.31% | 51.9m |
| T2T ($T_s$=50,$T_g$=30) [9] G+2Opt | 7.76 | 0.07% | 28.6m | 16.68 | 0.82% | 6.5m | 23.44 | 1.40% | 19.7m |
| T2T ($T_s$=50,$T_g$=30) [9] S+2Opt | 7.76 | 0.02% | 45.1m | 16.63 | 0.48% | 19.7m | 23.37 | 1.07% | 51.1m |
| Fast T2T ($T_s$=5,$T_g$=5) [24] G+2Opt | 7.76 | 0.03% | 4.2m | 16.61 | 0.39% | 2.2m | 23.25 | 0.58% | 8.6m |
| Fast T2T ($T_s$=5,$T_g$=5) [24] S+2Opt | 7.76 | 0.01% | 8.3m | 16.58 | 0.21% | 6.9m | 23.22 | 0.42% | 18.3m |
| GenSCO (C=10) | 7.76 | 0.000% | 6s | 16.55 | 0.023% | 5s | 23.25 | 0.559% | 12s |
| GenSCO (C=20) | – | – | – | 16.55 | 0.020% | 9s | 23.21 | 0.390% | 23s |
| GenSCO (C=40) | – | – | – | 16.55 | 0.016% | 18s | 23.18 | 0.258% | 46s |
| GenSCO (C=10) 2Opt | 7.76 | 0.000% | 6s | 16.55 | 0.019% | 5s | 23.13 | 0.063% | 16s |
| GenSCO (C=20) 2Opt | – | – | – | 16.55 | 0.016% | 9s | 23.13 | 0.054% | 30s |
| GenSCO (C=40) 2Opt | – | – | – | 16.55 | 0.012% | 18s | 23.13 | 0.046% | 58s |
| GenSCO (C=80) 2Opt | – | – | – | 16.55 | 0.011% | 36s | 23.13 | 0.041% | 2.0m |
| GenSCO (C=160) 2Opt | – | – | – | 16.55 | 0.010% | 1.2m | 23.13 | 0.036% | 3.9m |

solution quality (0.011%). Although LKH3 exhibits a steeper scaling slope and can eventually surpass GenSCO at larger problem sizes, this work represents a significant milestone where a neural solver genuinely outperforms LKH with sufficient time. Notably, within the high-accuracy regime of 0.02% optimality gap, GenSCO achieves approximately 10× speedup compared to LKH3.

**Scaling Effect with Increased Computation.** Fig.2 (a) presents a comparison of scaling curves between GenSCO and the raw diffusion model as computation scales, with 2Opt as the post-processing technique. We scale previous diffusion models through denoising steps (DIFUSCO [21]) and gradient search steps (T2T [9]). The results show that while gradient search can overcome performance plateaus, its scaling effectiveness remains limited due to the costly inference steps. In contrast, thanks to its lightweight operators, GenSCO efficiently scales to hundreds of function evaluations within a short

Table 2: Generalization results. Objective and Optimality Gap metrics are reported.

| Training / Testing | | TSP-100 | TSP-500 | TSP-1000 |
|---|---|---|---|---|
| TSP50 | DIFUSCO ($T_s$=50) [21] | 5.70, 0.25% | 5.83, 2.55% | 5.84, 2.71% |
| | T2T ($T_s$=50,$T_g$=30) [9] | 5.70, 0.11% | 5.78, 1.60% | 5.75, 1.10% |
| | GenSCO (C=40) | **5.69, 0.00%** | **5.69, 0.05%** | **5.69, 0.08%** |
| TSP100 | DIFUSCO ($T_s$=50) [21] | 7.78, 0.23% | 8.03, 3.44% | 8.02, 3.31% |
| | T2T ($T_s$=50,$T_g$=30) [9] | 7.77, 0.08% | 7.95, 2.47% | 7.91, 1.96% |
| | GenSCO (C=40) | **7.76, 0.00%** | **7.85, 1.16%** | **7.87, 1.42%** |
| TSP500 | DIFUSCO ($T_s$=50) [21] | 17.05, 3.04% | 16.78, 1.40% | 16.86, 1.85% |
| | T2T ($T_s$=50,$T_g$=30) [9] | **16.92, 2.25%** | 16.68, 0.81% | 16.72, 1.00% |
| | GenSCO (C=40) | 16.99, 2.70% | **16.55, 0.01%** | **16.55, 0.02%** |
| TSP1K | DIFUSCO ($T_s$=50) [21] | 24.04, 3.98% | 23.65, 2.30% | 23.63, 2.21% |
| | T2T ($T_s$=50,$T_g$=30) [9] | **23.85, 3.16%** | 23.47, 1.51% | 23.41, 1.23% |
| | GenSCO (C=40) | 24.37, 5.40% | **23.14, 0.11%** | **23.13, 0.05%** |

time frame, maintaining high performance with a nearly linear improvement trend. Fig.2 (b) illustrates the performance variation when scaling denoising steps in a single generation round and scaling search cycles in GenSCO. Scaling search cycles results in more stable performance, overcoming the performance plateau that occurs when simply increasing inference steps.

Table 4: Results on MIS. G: Greedy, S: Sampling, TS: Tree Search, UL: Unsupervised Learning.

| Method | Type | RB-[200-300] | | | ER-[700-800] | | |
|---|---|---|---|---|---|---|---|
| | | Obj. | Gap | Time | Obj. | Gap | Time |
| KaMIS [87] | Heuristics | 20.10* | – | 1.4h | 44.87* | – | 52.1m |
| Gurobi [88] | Exact | 19.98 | 0.01% | 47.6m | 41.28 | 7.78% | 50.0m |
| Intel [83] | SL+G | – | – | – | 34.86 | 22.31% | 6.1m |
| DIMES [12] | RL+G | – | – | – | 38.24 | 14.78% | 6.1m |
| Intel [83] | SL+TS | 18.47 | 8.11% | 13.1m | 38.80 | 13.43% | 20.0m |
| DGL [85] | SL+TS | 17.36 | 13.61% | 12.8m | 37.26 | 16.96% | 22.7m |
| LwD [84] | RL+S | – | – | – | 41.17 | 8.25% | 6.3m |
| GFlowNets [23] | UL+S | 19.18 | 4.57% | 32s | 41.14 | 8.53% | 2.9m |
| DIFUSCO ($T_s$=100) [21] | SL+G | 18.52 | 7.81% | 16.1m | 37.03 | 18.53% | 5.5m |
| DIFUSCO ($T_s$=100) [21] | SL+S | 19.13 | 4.79% | 20.5m | 39.12 | 12.81% | 21.7m |
| T2T ($T_s$=50,$T_g$=30) [9] | SL+G | 18.98 | 5.49% | 21.0m | 39.81 | 11.28% | 7.1m |
| T2T ($T_s$=50,$T_g$=30) [9] | SL+S | 19.38 | 3.53% | 30.3m | 41.41 | 7.72% | 27.8m |
| Fast T2T ($T_s$=5,$T_g$=5) [24] | SL+G | 19.49 | 2.89% | 4.7m | 40.68 | 9.34% | 1.5m |
| Fast T2T ($T_s$=5,$T_g$=5) [24] | SL+S | 19.70 | 1.90% | 7.0m | 41.73 | 6.99% | 5.9m |
| GenSCO (C=500) | SL+G | 19.85 | 1.24% | 1.1m | 42.46 | 5.37% | 40s |
| GenSCO (C=1k) | SL+G | 19.91 | 0.96% | 2.2m | 42.91 | 4.38% | 1.3m |
| GenSCO (C=2k) | SL+G | 19.97 | 0.64% | 4.4m | 43.33 | 3.44% | 2.7m |
| GenSCO (C=4k) | SL+G | 20.01 | 0.43% | 8.8m | 43.77 | 2.46% | 5.3m |
| GenSCO (C=8k) | SL+G | 20.04 | 0.29% | 17.6m | 44.13 | 1.66% | 10.7m |
| GenSCO (C=16k) | SL+G | 20.06 | 0.22% | 35.2m | 44.41 | 1.03% | 21.4m |

**Cross-Scale Generalization.** We train the model on a specific problem scale and then evaluate it on all problem scales. Table 2 presents the generalization results of GenSCO compared with diffusion counterparts. We discover that generalization performs well between large-scale datasets (TSP-500 and 1000) and also between small-scale datasets (TSP-50 and 100). However, generalization degrades across data distributions with large differences, unless the test set is a simpler dataset like TSP-50. Nonetheless, GenSCO still exhibits superior performance compared to prior baselines.

**Training with Suboptimal Supervision.** We investigate the performance of GenSCO trained with worse-quality data produced by the 2Opt heuristic (with 1.9% gap on TSP-100 and 7.0% gap on TSP-500). Table 3 shows that even though the model is trained on worse-quality solutions, which results in a less reliable solution refinement mapping, the iterative search cycles can still effectively explore the solution space, thereby ultimately yielding performance that surpasses the original supervision source. We further employ the trained GenSCO to relabel a small subset (1/75) of the dataset, then iteratively fine-tune subsequent models (GenSCO² and GenSCO³) to verify the capacity of self-improving and gradually elevating the supervision quality.

## 5.2 Experiments for MIS

**Datasets.** Two datasets are tested for the MIS problem following [83, 84, 85, 12, 21], including RB graphs [23] and Erdős–Rényi (ER) graphs [86]. For RB graphs, we randomly sample 200 to 300 vertices uniformly to generate graph instances. For ER graphs, we construct random graphs with 700 to 800 nodes, setting the pairwise connection probability as 0.15.

Table 3: Performance of models trained with lower-quality supervision.

| Method | TSP-100 | | TSP-500 | |
|---|---|---|---|---|
| | Obj. | Gap | Obj. | Gap |
| 2Opt | 7.905 | 1.920% | 17.703 | 6.994% |
| GenSCO | 7.757 | 0.014% | 16.823 | 1.676% |
| GenSCO² | 7.756 | 0.007% | 16.699 | 0.927% |
| GenSCO³ | 7.756 | 0.005% | 16.642 | 0.583% |

**Metrics.** 1) Objective: the average size of the solutions w.r.t. the corresponding instances; 2) Drop: the relative performance drop w.r.t. size compared to the optimal solution or the reference solution; 3) Time: the computational time required to solve all the test instances.

**Model Setting.** We use 3 denoising steps in a single round of flow generation and run the search operator without parallelization. The number of search cycles, denoted as $C$, is treated as a hyperparameter to control the extent of exploration. The configurations of the diffusion baselines follow those in TSP.

**Main Results.** Table 4 demonstrates the effectiveness of GenSCO, showcasing both performance advantages and stable scaling behavior compared to the raw diffusion model. To ensure a fair comparison with neural solvers, Gurobi's solving time is limited, and thus it does not reach optimality.

Table 5: Results on MCl. G: Greedy, S: Sampling, UL: Unsupervised Learning.

| Method | Type | RB-[200-300] | | | RB-[800-1200] | | |
|---|---|---|---|---|---|---|---|
| | | Obj. | Gap | Time | Obj. | Gap | Time |
| Gurobi [88] | Exact | 19.08* | 0.00% | 7.5m | 40.18 | 0.00% | 38.4h |
| Meta-EGN [89] | UL | 17.51 | 8.30% | 2.3m | 33.79 | 15.49% | 4.5m |
| DiffUCO [16] | UL | 16.21 | 12.53% | 11.8m | – | – | – |
| COExpander [53] bs | SL+S | 19.00 | 0.50% | 5.4m | 39.06 | 2.99% | 53.0m |
| COExpander [53] more steps | SL+S | 18.98 | 0.68% | 30s | 39.88 | 0.90% | 3.1m |
| GenSCO (C=500) | SL+G | 19.08 | 0.01% | 1.1m | 40.22 | -0.10% | 6.9m |
| GenSCO (C=1k) | SL+G | 19.08 | 0.00% | 2.2m | 40.23 | -0.13% | 13.8m |

GenSCO (C=500) achieves strong results within just 1.1 minutes on the RB dataset and 40 seconds on the ER dataset, reaching optimality gaps of 1.24% and 5.37%, respectively. This corresponds to a $3.9\times$ reduction in gap with an $18.6\times$ speedup on RB, and a $2.4\times$ gap reduction with a $32.6\times$ speedup on ER, compared to the raw diffusion model baseline DIFUSCO [21] when scaling inference steps. Moreover, as the number of search cycles increases, GenSCO consistently delivers performance gains. With comparable solving time, it reduces the optimality gap from 4.79% to 0.29% ($16.5\times$ improvement) on RB, and from 12.81% to 1.03% ($12.4\times$ improvement) on ER, compared to DIFUSCO.

## 5.3 Experiments for MCl

**Datasets.** Two datasets, RB-Small and RB-Large, are tested for the MCl problem following [53, 23, 16]. The RB-Small dataset involves graph instances with 200 to 300 uniformly sampled vertices, and the RB-Large dataset involves graph instances with 800 to 1200 uniformly sampled vertices.

**Metrics.** 1) Objective: the average size of the cliques w.r.t. the corresponding instances; 2) Drop: the relative performance drop w.r.t. size compared to the optimal solution or the reference solution; 3) Time: the computational time required to solve all the test instances.

**Model Setting.** We use 3 denoising steps in a single round of flow generation and run the search operator without parallelization. The number of search cycles, denoted as $C$, is treated as a hyperparameter to control the extent of exploration. The configurations of the diffusion baselines follow those in TSP and MIS.

**Main Results.** Table 5 presents the significant performance advantages of GenSCO on the MCl problem. To ensure a fair comparison with neural solvers, Gurobi's solving time is limited, and thus it does not reach optimality. GenSCO achieves strong results with the solution quality gap of 0.00% on RB-[200-300] and -0.13% on RB-[800-1200] compared to Gurobi. This exceptional solution quality is delivered with remarkable efficiency. GenSCO can exceed Gurobi (-0.10%) with just 6.9 minutes ($334\times$ speedup), which stands in stark contrast to the 38.4 hours needed by Gurobi to reach such solution quality. Compared to other learning-based solvers, this outcome marks a step-change, order-of-magnitude improvement. From the perspective of the gap metric, it has reduced the learning-based solvers' performance from 0.90% to -0.13%.

## 6 Conclusion

This work presents a novel perspective on diffusion-based solvers by prioritizing inference-time cost-efficiency through scalable, search-driven optimization rather than focusing solely on maximizing single-round generation quality. We redesign both the diffusion architecture and the solving pipeline by introducing a solution-to-solution rectified flow model alongside a controllable search operator, enabling systematic exploration of the solution space via iterative refinement cycles. Extensive experiments demonstrate that GenSCO delivers performance improvements by orders of magnitude over previous state-of-the-art neural methods. Notably, GenSCO even achieves significant speedups compared to the state-of-the-art classic mathematical solver LKH3, delivering a $141\times$ speedup to reach 0.000% optimality gap on TSP-100, and approximately a $10\times$ speedup to reach 0.02% on TSP-500. These results potentially mark a milestone in neural combinatorial optimization, showcasing the potential of scalable neural solvers for high-accuracy and high-speed problem solving and paving the way for future advances in combinatorial optimization through the lens of AI.

## Acknowledgements

This work was partly supported by NSFC (92370201).

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

# Appendix

## A    Broad Impacts, Limitations and Future Work

This paper explores test-time scaling for generative neural solvers, demonstrating that generative models designed specifically with search-oriented scaling can significantly improve performance. Experiments show that the proposed algorithm achieves orders-of-magnitude improvements compared to state-of-the-art machine learning solvers, and also exhibits notable advantages over LKH on TSP instances of size 500 and below. The algorithm reveals the great potential of neural network solvers to surpass the decades-refined mathematical heuristic algorithms like LKH on some classical problems that have been thoroughly studied.

Current limitations include that in larger-scale experiments, we observed that given more time, the improvement rate of GenSCO slows down, and eventually the improvement rate of LKH surpasses that of GenSCO. Maintaining an effective, continued search on larger-scale data is a target for future exploration. On the other hand, switching between neural network solvers and traditional solvers in a two-stage manner is a possible direction for further research. In generalization experiments, we found that the model suffers from performance degradation on certain hard TSPLIB instances, which impacts the average performance. Improving the robustness of neural network solvers is also a future direction.

## B    Claim for the Utilization of Parallel GenSCO Operators

We note that the core logic of our methodology is: 1) to explore a new solving paradigm (generation as a search operator) that offers better scaling effects, and 2) to maximize the use of the proposed operator to significantly boost solving performance. In this design, the neural network's role is lightweight, with a low number of inference steps per generation, which is designed to reallocate the computation to more search cycles. Compared to previous methods that rely on the accuracy of single-round heavy generation, GenSCO's single neural network inference has a low GPU computational density. Parallelism is necessary to ensure GenSCO fully utilizes the available GPU power, making the computational cost comparable to other methods that rely on "heavy" networks for a single model inference.

Thus, the results reported in the main experiments reflect the parallel execution of 8 GenSCO operators. Here, Table 6 shows results for 1, 2, and 4 parallel operators to show the effect of varying parallel runs. As seen, even with the num_runs=1 setting, GenSCO still significantly outperforms other baselines (without full GPU utilization).

Table 6: Ablation studies of GenSCO with different parallel runs on TSP-500 and TSP-1000.

| Method | Runs | TSP-500 | | | TSP-1000 | | |
|---|---|---|---|---|---|---|---|
| | | Obj. | Gap (%) | Time (s) | Obj. | Gap (%) | Time (s) |
| GenSCO (C=40) | 1 | 16.550 | 0.028 | 2.408 | 23.137 | 0.083 | 7.609 |
| | 2 | 16.549 | 0.021 | 4.383 | 23.133 | 0.065 | 12.372 |
| | 4 | 16.548 | 0.016 | 9.248 | 23.130 | 0.053 | 24.164 |
| | 8 | 16.548 | 0.013 | 17.974 | 23.129 | 0.047 | 47.468 |
| GenSCO (C=80) | 1 | 16.549 | 0.021 | 4.365 | 23.134 | 0.067 | 14.012 |
| | 2 | 16.549 | 0.016 | 8.333 | 23.131 | 0.055 | 23.834 |
| | 4 | 16.548 | 0.012 | 17.904 | 23.129 | 0.047 | 47.327 |
| | 8 | 16.548 | 0.011 | 35.541 | 23.127 | 0.040 | 93.989 |
| GenSCO (C=160) | 1 | 16.548 | 0.016 | 8.226 | 23.131 | 0.056 | 27.741 |
| | 2 | 16.548 | 0.013 | 16.160 | 23.129 | 0.047 | 46.770 |
| | 4 | 16.548 | 0.011 | 35.375 | 23.128 | 0.041 | 93.904 |
| | 8 | 16.548 | 0.010 | 70.923 | 23.126 | 0.034 | 187.634 |

# C   Supplementary Experiments

## C.1   Results on TSP Real-World Data

**Results on TSPLIB 50-200.** We evaluate our model trained with random 100-node problems on real-world TSPLIB instances with 50-200 nodes. The compared baselines include DIFUSCO [21], T2T [9], Fast T2T [24], and baselines listed in [91]'s Table 3. The hyperparameter settings of the compared baselines are: DIFUSCO: $T_s$=50; T2T: $T_s$=50 and $T_g$=30; GenSCO (w/o GS): $T_s$=10; Fast T2T (w/ GS): $T_s$=10 and $T_g$=10; GenSCO: we adopt 4 inference steps in one generation round, 200 search cycles, and 16 operators in parallel. The diffusion-based methods are compared in the same settings with greedy decoding and Two-Opt post-processing. For each instance, we normalize the coordinates to [0,1]. The results are presented in Table 7.

**Results on TSPLIB 200-1000.** We also supplement the results (optimality drop) of diffusion-based baselines and GenSCO on large-scale TSPLIB benchmark instances with 200-1000 nodes. The models are trained on TSP-500 and inference with greedy decoding and Two-Opt post-processing. For each instance, we normalize the coordinates to [0,1]. The results are presented in Table 8.

Table 7: Solution quality for methods trained on random 100-node problems and evaluated on **TSPLIB instances** with 50-200 nodes. * denotes results quoted from previous works [91].

| INSTANCES | AM* | GCN* | Learn2OPT* | GNNGLS* | DIFUSCO | T2T | Fast T2T | GenSCO |
|---|---|---|---|---|---|---|---|---|
| eil51 | 16.767% | 40.025% | 1.725% | 1.529% | 2.82% | 0.14% | 0.00% | 0.000% |
| berlin52 | 4.169% | 33.225% | 0.449% | 0.142% | 0.00% | 0.00% | 0.00% | 0.000% |
| st70 | 1.737% | 24.785% | 0.040% | 0.764% | 0.00% | 0.00% | 0.00% | 0.000% |
| eil76 | 1.992% | 27.411% | 0.096% | 0.163% | 0.34% | 0.00% | 0.00% | 0.000% |
| pr76 | 0.816% | 27.793% | 1.228% | 0.039% | 1.12% | 0.40% | 0.00% | 0.000% |
| rat99 | 2.645% | 17.633% | 0.123% | 0.550% | 0.09% | 0.09% | 0.00% | 0.000% |
| kroA100 | 4.017% | 28.828% | 18.313% | 0.728% | 0.10% | 0.00% | 0.00% | 0.000% |
| kroB100 | 5.142% | 34.686% | 1.119% | 0.147% | 2.29% | 0.74% | 0.65% | 0.000% |
| kroC100 | 0.972% | 35.506% | 0.349% | 1.571% | 0.00% | 0.00% | 0.00% | 0.000% |
| kroD100 | 2.717% | 38.018% | 0.866% | 0.572% | 0.07% | 0.00% | 0.00% | 0.000% |
| kroE100 | 1.470% | 26.589% | 1.832% | 1.216% | 3.83% | 0.27% | 0.00% | 0.000% |
| rd100 | 3.407% | 50.432% | 1.725% | 0.003% | 0.08% | 0.00% | 0.00% | 0.000% |
| eil101 | 2.994% | 26.701% | 0.387% | 1.529% | 0.03% | 0.00% | 0.00% | 0.000% |
| lin105 | 1.739% | 34.902% | 1.867% | 0.606% | 0.00% | 0.00% | 0.00% | 0.000% |
| pr107 | 3.933% | 80.564% | 0.898% | 0.439% | 0.91% | 0.61% | 0.62% | 0.000% |
| pr124 | 3.677% | 70.146% | 10.322% | 0.755% | 1.02% | 0.60% | 0.08% | 0.000% |
| bier127 | 5.908% | 45.561% | 3.044% | 1.948% | 0.94% | 0.54% | 1.50% | 0.000% |
| ch130 | 3.182% | 39.090% | 0.709% | 3.519% | 0.29% | 0.06% | 0.00% | 0.000% |
| pr136 | 5.064% | 58.673% | 0.000% | 3.387% | 0.19% | 0.10% | 0.01% | 0.000% |
| pr144 | 7.641% | 55.837% | 1.526% | 3.581% | 0.80% | 0.50% | 0.39% | 0.000% |
| ch150 | 4.584% | 49.743% | 0.312% | 2.113% | 0.57% | 0.49% | 0.00% | 0.000% |
| kroA150 | 3.784% | 45.411% | 0.724% | 2.984% | 0.34% | 0.14% | 0.00% | 0.000% |
| kroB150 | 2.437% | 56.745% | 0.886% | 3.258% | 0.30% | 0.00% | 0.07% | 0.340% |
| pr152 | 7.494% | 33.925% | 0.029% | 3.119% | 1.69% | 0.83% | 0.19% | 0.187% |
| u159 | 7.551% | 38.338% | 0.054% | 1.020% | 0.82% | 0.00% | 0.00% | 0.000% |
| rat195 | 6.893% | 24.968% | 0.743% | 1.666% | 1.48% | 1.27% | 0.79% | 0.194% |
| d198 | 373.020% | 62.351% | 0.522% | 4.772% | 3.32% | 1.97% | 0.86% | 0.751% |
| kroA200 | 7.106% | 40.885% | 1.441% | 2.029% | 2.28% | 0.57% | 0.49% | 0.160% |
| kroB200 | 8.541% | 43.643% | 2.064% | 2.589% | 2.35% | 0.92% | 2.50% | 0.098% |
| **Mean** | 16.767% | 40.025% | 1.725% | 1.529% | 0.97% | 0.35% | 0.28% | 0.061% |

## C.2   Ablation Study on the Flow Modeling

Table 9 compares the flow model and the consistency model (from the state-of-the-art Fast T2T [24]) on TSP-500 under the same implementation and number of forward function evaluations (NFE). We treat the consistency model as an alternative diffusion model variant and enhance its competitiveness by applying more 2-opt cycles, though at the cost of increased runtime. The results demonstrate the effectiveness of our proposed flow model over the previous consistency model design.

Table 8: Solution quality for methods trained on random 500-node problems and evaluated on **TSPLIB instances** with 200-1000 nodes.

| INSTANCES | DIFUSCO | T2T | Fast T2T | GenSCO |
|---|---|---|---|---|
| a280 | 1.39% | 1.39% | 0.10% | 0.000% |
| d493 | 1.81% | 1.81% | 1.43% | 3.872% |
| d657 | 4.86% | 2.40% | 0.64% | 0.457% |
| fl417 | 3.30% | 3.30% | 2.01% | 0.596% |
| gil262 | 2.18% | 0.96% | 0.18% | 0.000% |
| lin318 | 2.95% | 1.73% | 1.21% | 0.025% |
| linhp318 | 2.17% | 1.11% | 0.78% | 0.025% |
| p654 | 7.49% | 1.19% | 1.67% | 1.023% |
| pcb442 | 2.59% | 1.70% | 0.61% | 0.015% |
| pr226 | 4.22% | 0.84% | 0.34% | 0.211% |
| pr264 | 0.92% | 0.92% | 0.73% | 0.000% |
| pr299 | 1.46% | 1.46% | 1.40% | 0.011% |
| pr439 | 2.73% | 1.63% | 0.50% | 0.106% |
| rat575 | 2.32% | 1.29% | 1.43% | 0.000% |
| rat783 | 3.04% | 1.88% | 1.03% | 0.068% |
| rd400 | 1.18% | 0.44% | 0.08% | 0.030% |
| ts225 | 4.95% | 2.24% | 1.37% | 0.517% |
| tsp225 | 3.25% | 1.69% | 0.81% | 0.000% |
| u574 | 2.50% | 1.85% | 0.94% | 0.109% |
| u724 | 2.05% | 2.05% | 1.41% | 0.040% |
| **Mean** | 2.87% | 1.59% | 0.93% | 0.355% |

Table 9: Ablation studies of diffusion modeling design choices on TSP-500.

| Method | NFE=40 | | | NFE=80 | | | NFE=160 | | |
|---|---|---|---|---|---|---|---|---|---|
| | Obj. | Gap | Time | Obj. | Gap | Time | Obj. | Gap | Time |
| Consistency [24] | 16.598 | 0.313% | 7s | 16.590 | 0.265% | 14s | 16.584 | 0.233% | 28s |
| Consistency [24] 2Opt | 16.550 | 0.028% | 23s | 16.549 | 0.022% | 46s | 16.549 | 0.021% | 1m33s |
| Flow | 16.550 | 0.023% | 5s | 16.549 | 0.020% | 9s | 16.548 | 0.016% | 18s |
| Flow 2Opt | 16.549 | 0.019% | 5s | 16.548 | 0.016% | 9s | 16.548 | 0.012% | 18s |

## C.3 Ablation Study on the Architecture Choices

Table 10 compares different architecture choices within GenSCO to show the effectiveness of the proposed Transformer architecture compared to the classic Graph Convolutional Network (GCN). We reimplement the GCN following the specific settings in DIFUSCO [21] and Fast T2T [24] as an alternative backbone for the flow model. The GCN implementation takes much more runtime for the same iterations, and the efficiency of the proposed Transformer architecture allows for the maximization of the scaling benefits.

Table 10: Ablation studies of architecture choices on TSP-500 and TSP-1000.

| Method | TSP-500 | | | TSP-1000 | | |
|---|---|---|---|---|---|---|
| | Obj. | Gap | Time | Obj. | Gap | Time |
| GenSCO (C=40) with GCN | 16.556 | 0.060% | 1m45s | 23.157 | 0.167% | 6m50s |
| GenSCO (C=40) with Transformer | 16.549 | 0.019% | 5s | 23.133 | 0.063% | 16s |
| GenSCO (C=80) with GCN | 16.553 | 0.045% | 3m25s | 23.147 | 0.127% | 13m40s |
| GenSCO (C=80) with Transformer | 16.548 | 0.016% | 9s | 23.131 | 0.054% | 30s |
| GenSCO (C=160) with GCN | 16.552 | 0.037% | 6m50s | 23.140 | 0.097% | 27m20s |
| GenSCO (C=160) with Transformer | 16.548 | 0.012% | 18s | 23.129 | 0.046% | 58s |

## C.4 Ablation Study on the Augmentation Technique

In the implementation, at each search cycle, we apply a symmetric transformation to the instance, such as rotating the coordinates. This operation increases the diversity of the model prediction during

the intermediate process, ensuring the continuity of exploration ability throughout the search. We conduct an ablation study on this trick in Table 11.

Table 11: Ablation study of the augmentation technique on TSP-100 and 500.

| Method | TSP-100 (1280 inst.) | | | TSP-500 (128 inst.) | | |
|---|---|---|---|---|---|---|
| | Obj. | Gap | Time | Obj. | Gap | Time |
| GenSCO (C=10) w/ aug. | 7.76 | 0.000% | 6s | 16.55 | 0.023% | 5s |
| w/o aug. | 7.76 | 0.004% | 6s | 16.55 | 0.047% | 5s |
| GenSCO (C=20) w/ aug. | 7.76 | 0.000% | 12s | 16.55 | 0.020% | 9s |
| w/o aug. | 7.76 | 0.003% | 12s | 16.55 | 0.040% | 9s |
| GenSCO (C=40) w/ aug. | 7.76 | 0.000% | 24s | 16.55 | 0.016% | 18s |
| w/o aug. | 7.76 | 0.003% | 24s | 16.55 | 0.031% | 18s |
| GenSCO (C=10) 2Opt w/ aug. | 7.76 | 0.000% | 6s | 16.55 | 0.019% | 5s |
| w/o aug. | 7.76 | 0.003% | 6s | 16.55 | 0.037% | 5s |
| GenSCO (C=20) 2Opt w/ aug. | 7.76 | 0.000% | 12s | 16.55 | 0.016% | 9s |
| w/o aug. | 7.76 | 0.003% | 12s | 16.55 | 0.032% | 9s |
| GenSCO (C=40) 2Opt w/ aug. | 7.76 | 0.000% | 24s | 16.55 | 0.012% | 18s |
| w/o aug. | 7.76 | 0.003% | 24s | 16.55 | 0.026% | 18s |

## C.5 Ablation Study on the Disruption Operator

In the implementation, at each search cycle, we apply a disruption operator to the obtained solutions to escape from local optima and achieve continuous improvement. Table 12 shows the results of GenSCO with and without the disruption operator. Without the disruption operator, we observe varying degrees of performance degradation, especially on TSP500, where the gains from increasing search cycles show significant stagnation.

Table 12: Ablation study of the disruption operator on TSP-100 and 500.

| Method | TSP-100 | | TSP-500 | |
|---|---|---|---|---|
| | Obj. | Gap | Obj. | Gap |
| GenSCO (C=40) w/ disruption | 16.549 | 0.019% | 23.133 | 0.063% |
| w/o disruption | 16.551 | 0.033% | 23.137 | 0.083% |
| GenSCO (C=80) w/ disruption | 16.548 | 0.016% | 23.131 | 0.054% |
| w/o disruption | 16.551 | 0.032% | 23.134 | 0.067% |
| GenSCO (C=160) w/ disruption | 16.548 | 0.012% | 23.129 | 0.046% |
| w/o disruption | 16.551 | 0.032% | 23.131 | 0.056% |

## C.6 Ablation Study on the Number of Sampling Steps

Table 13 shows the effect of varying the number of sampling steps for GenSCO. We observe that increasing the number of sampling steps generally improved performance for GenSCO with different numbers of cycles, though the gains diminished in later stages. We ultimately chose $T = 4$ and allocated more computational resources to scaling the number of cycles. .

Table 13: Ablation studies of the number of sampling steps on TSP-500.

| T | C=40 | | | C=80 | | | C=160 | | |
|---|---|---|---|---|---|---|---|---|---|
| | Obj. | Gap | Time | Obj. | Gap | Time | Obj. | Gap | Time |
| 1 | 16.561 | 0.091% | 8.06s | 16.558 | 0.072% | 15.54s | 16.556 | 0.059% | 31.47s |
| 2 | 16.549 | 0.018% | 8.34s | 16.549 | 0.017% | 16.33s | 16.548 | 0.015% | 32.15s |
| 4 | 16.548 | 0.013% | 18.38s | 16.548 | 0.011% | 35.56s | 16.548 | 0.010% | 70.62s |
| 6 | 16.548 | 0.013% | 26.88s | 16.548 | 0.011% | 53.35s | 16.547 | 0.009% | 106.26s |
| 8 | 16.548 | 0.012% | 35.60s | 16.547 | 0.010% | 70.66s | 16.547 | 0.009% | 140.92s |

## C.7 Visualization of the Generation

Fig. 3 shows the interpolation achieved by neural predictions between a suboptimal solution and the optimal one, which is a typical process in inference. In the inference process, the search operator first disrupts the solution with random local search operators and then conducts the flow sampling process. As corresponding to this specific process, Fig. 5 presents the interpolation achieved by neural predictions between a disrupted suboptimal solution and the optimal one. Fig. 4 shows Flow-based interpolation between an intermediate state (lying between a suboptimal and the optimal solution) and the optimal solution, which is a typical process in training.

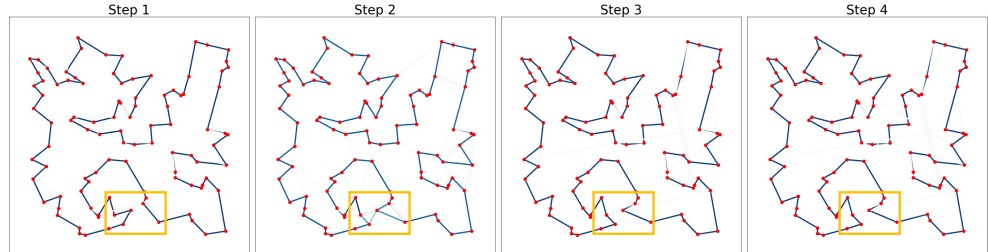

Figure 3: Flow-based interpolation between a suboptimal solution and the optimal one.

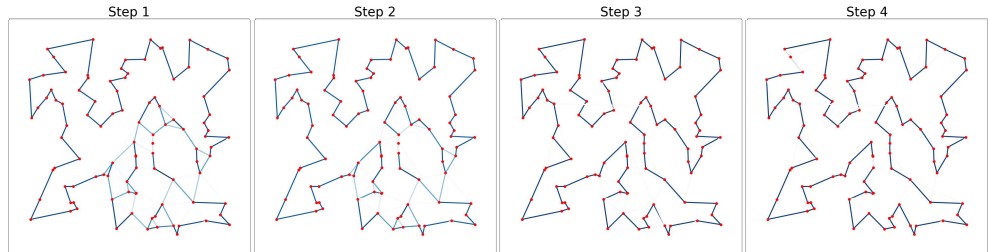

Figure 4: Flow-based interpolation achieved by neural predictions between a disrupted suboptimal solution and the optimal one.

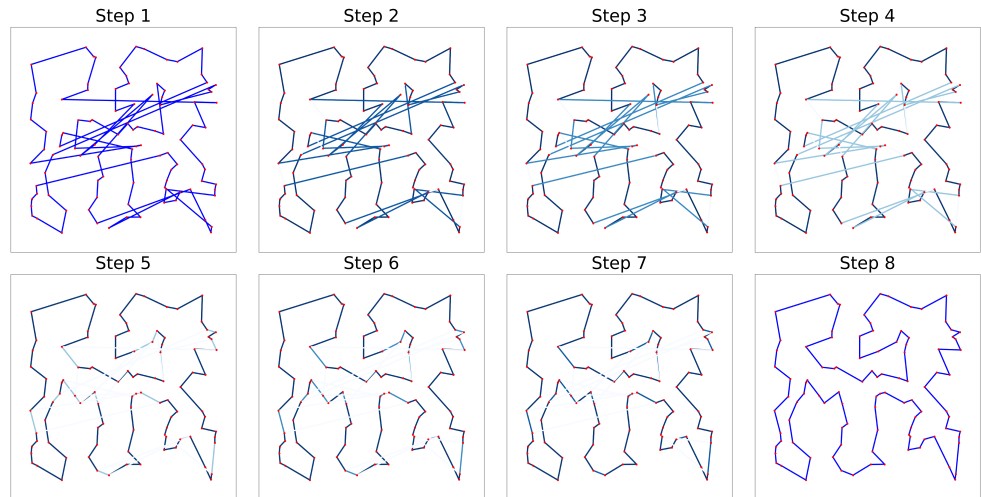

Figure 5: Flow-based interpolation between a suboptimal solution and the optimal one.

# D   Experimental Details

## D.1   Computational Resources.

All test evaluations are performed on a single GPU of the NVIDIA RTX 4090. The training costs of GenSCO on TSP-100/500/1000 are <= 2 days, <= 1 day, and <= 1 day on a GTX 4090 24G. For comparison, DIFUSCO [21] takes 8.6 days, 2.7 days, and 5.1 days for training TSP-100/500/1000 on an A100 GPU, respectively.

## D.2   Design Choices and Hyperparameters

To ensure fair and thorough evaluation across problem scales, we run multiple experiments on TSP-100, TSP-500, and TSP-1000 using certain fixed hyperparameters and random seeds.

**Two-opt steps** are scaled with problem size: we set `two_opt_steps = n/50`, where $n$ is the number of nodes in the TSP instance. This results in 2 steps for TSP-100, 10 steps for TSP-500, and 20 steps for TSP-1000.

**Randomized 2-opt search** is applied with a range proportional to problem size, using `random_two_opt_steps_range = [n/4, 3n/4]`.

**Robustness** is ensured by repearing each setting with 5 different random seeds, and giving the final result as the average across these five runs.

**Adjacent matrix heatmap filtering** selects top-k candidate edges using `topk = 20000` for TSP-500 and TSP-1000, and 5000 for TSP-100.

## D.3   Datasets.

The reference solutions for TSP-100/500/1000 are labeled by the Concorde exact solver [79]. The test set for TSP-50/100 is taken from [1, 8] with 1280 instances, and the test set for TSP-500/1000 is from [19] with 128 instances for the fair comparison.

The reference solutions for both RB graphs and ER graphs are labeled with KaMIS [87]. For RB graphs, we randomly generate 90000 instances for the training set and 500 instances for the test set. For ER graphs, we randomly generate 163840 instances for the training set, and the test is from [12].

## D.4   Baseline Settings

The generative baselines are compared in the same running settings, while the results of other baselines are quoted from the best achieved results of their original papers.

### D.4.1   TSP Benchmarks

In the evaluation of TSP-50/100/500, we compare our proposed GenSCO against 15 baseline methods. These baselines include one exact solver, i.e., Concorde [79]; one heuristic solver, i.e., LKH-3 [80]; nine non-generative learning-based solvers, i.e., AM [1], POMO [2], POMO+EAS [81] GCN [8], DIMES [12], LEHD [20], BQ-NCO [22], GLOP [39], and UDC [40], COExpander [53]; and three generative learning-based solvers, i.e., DIFUSCO [21], T2T [9], and Fast T2T [24]. These learning-based methods can be further categorized into supervised learning (SL) and reinforcement learning (RL). Post-processing techniques employed encompass greedy decoding (Grdy, G), multiple sampling (S), 2Opt refinement (2Opt), beam search (BS), active search (AS), and combinations thereof. We cap the number of inference steps for DIFUSCO at 100. For T2T, we fix the number of inference steps and guided search steps at 50 and 30, respectively. For Fast T2T, we fix the number of inference steps and guided search steps at 5 and 5, respectively.

### D.4.2   MIS Benchmarks

We assess our method on two distinct benchmarks: RB-[200-300] and ER-[700-800]. Across both benchmarks, we compare the performance of GenSCO against one exact solver, Gurobi [88], one heuristic solver, KaMIS [87], and 5 learning-based frameworks: Intel [83], DGL [83], LwD [84], DIMES [12], GFlowNets [23], DIFUSCO [21], T2T [9], and Fast T2T [24]. These learning-based

methods can be further categorized into supervised learning (SL), reinforcement learning (RL), and unsupervised learning (UL). Post-processing strategies encompass greedy decoding (G), multiple sampling (S), and tree search (TS). Specifically, on both benchmarks, we set the number of inference steps at 100 for DIFUSCO. For T2T, we set the number of inference steps and guided search steps at 50 and 30, respectively. For Fast T2T, we set the number of inference steps and guided search steps at 5 and 5, respectively.

### D.4.3 MCl Benchmarks

We assess our method on two distinct benchmarks: RB-[200-300] and ER-[800-1200]. Across both benchmarks, we compare the performance of GenSCO against one exact solver, Gurobi [88], and three learning-based frameworks: Meta-EGN [89], DiffUCO [16], COExpander [53]. These learning-based methods can be further categorized into supervised learning (SL) and unsupervised learning (UL).

## E   Network Architecture Details

### E.1   Network Architecture for TSP

**Input Layer.** The model takes raw features with dimensionality of 2 as input, typically representing coordinate data. These raw inputs are first projected into a higher-dimensional embedding space via a learnable linear layer. This initial projection maps the input to the encoder's embedding dimension (default 256).

**Attention Layers.** The architecture comprises an encoder-decoder Transformer. The encoder has 16 Transformer layers with 8 attention heads, 256-dim hidden states. Feed-forward layers use SwiGLU activations. The decoder has 6 layers with similar settings.

**Output Layer.** The output is generated through an MLP block that projects the decoder embeddings back into the model's embedding space. The logit representing the connection between two different nodes with transformed embeddings $h_1$ and $h_2$ is computed via inner product $\langle h_1, h_2 \rangle$.

### E.2   Network Architecture for MIS and MCl

**Input Layer.** The model accepts adjacency matrices representing graph connectivity. Initial node features are sampled from a standard normal distribution and linearly projected into the encoder embedding space. This initial embedding dimension (default 256) sets the feature size for subsequent Transformer encoding.

**Attention Layers.** We omit the encoder for MIS. The decoder uses 256-dimensional embeddings and 4 attention heads, with 12 and 8 layers employed for RB and ER graphs, respectively.

**Output Layer.** The model employs a sigmoid output head, which applies a sigmoid activation to scalar logits produced by an MLP to generate predictions.

