# OpenReview forum: "Generation as Search Operator for Test-Time Scaling of Diffusion-based Combinatorial Optimization"
_NeurIPS.cc/2025/Conference — NeurIPS 2025 poster_

### Official Review · Reviewer_vfnv · 2025-06-17

**Clarity:** 2
**Significance:** 2
**Originality:** 3
**Rating:** 4
**Confidence:** 3

**Summary:**

This paper introduces GenSCO to solve the inference-time cost‐efficiency bottleneck of diffusion models on combinatorial optimization tasks. GenSCO treats each generation pass as a search operator and interleaves local search disruptions with learned diffusion sampling in lightweight refinement cycles. A rectified flow model is trained via a bias‐corrected objective to perform solution‐to‐solution refinements, enabling controllable exploration of the near‐optimal solution space. The key insight is to leverage the learned high-quality solution distribution across multiple fast, iterative steps rather than relying on a single expensive generation. Experiments on TSP-100 and TSP-500 benchmarks show that GenSCO outperforms prior neural methods by orders of magnitude and achieves up to 141× speedup to zero optimality gap on TSP-100 and ~10× speedup to 0.02 % gap on TSP-500, demonstrating both enhanced solution quality and practical scalability.

**Questions:**

1. The motivation for using a rectified flow model for generation is underdeveloped—why choose rectified flow instead of directly using an autoregressive model?
2. Please provide a detailed comparison of model parameter counts to ensure the comparisons are fair.
3. Ablation studies on different modules are insufficient; there is no empirical evidence showing how the disruption operator, number of sampling steps, and post-processing affect performance.
4. How does this scaling approach differ from the traditional diffusion model’s iterative re-noising and de-noising process? Does it offer any clear performance advantages?
5. Can you provide empirical evidence demonstrating the advantages of solution-to-solution denoising over the original noise-to-solution denoising?
6. In Lines 153–157, could the proposed disruption method generate invalid solutions, and if so, how would that impact overall performance?

**Ethical Concerns:**

["NO or VERY MINOR ethics concerns only"]

**Final Justification:**

Thank you to the authors for the detailed rebuttal and for providing substantial new experiments. Your responses were very effective and have resolved my primary concerns. I have raised my score accordingly.

My main initial reservations were about the method's generality, the motivation for using rectified flow, and the lack of detailed ablation studies. The rebuttal addressed these points convincingly:

Generality and Scalability: The new experiments on the Max Clique problem were a welcome and compelling addition. Seeing the method not only adapt to a different CO task but also outperform a strong commercial solver like Gurobi significantly strengthens the paper's claims about its broader applicability.

Methodological Justification: Your clarification on using rectified flow to enable a more controllable, efficient, and scalable search process makes perfect sense. The comparison to traditional re-noising methods highlighted the advantages of your solution-to-solution refinement approach.

Ablation Studies: The new ablations on the disruption operator and the number of sampling steps provided the necessary evidence to understand the contribution of each component. It's now clear that the disruption operator is crucial for escaping local optima, especially on larger problem instances.

After the rebuttal, I feel the paper is in a much stronger position. There are no major unresolved issues.

**Limitations:**

yes

**Paper Formatting Concerns:**

No obvious format issues.

**Quality:**

3

**Strengths And Weaknesses:**

### Strengths

1. **Quality:** The writing and figures are complete and clear; method details are presented directly and are easy to understand.
2. **Significance:** The paper tackles an important combinatorial optimization problem and provides a comprehensive deep-learning–based approach.
3. **Originality:** Introducing the number of search cycles as a new scaling dimension—and demonstrating its effectiveness in Figure 2—along with using rectified flow in the relaxed solution space, improves cost-efficiency.
4. **Quality / Originality:** The authors systematically redesign both the diffusion architecture and the solving pipeline specifically for search-driven optimization.
5. **Significance:** Experimental validation on TSP and MIS datasets demonstrates the method’s practical effectiveness.

### Weaknesses

1. **Originality / Quality:** The motivation for using a rectified flow model for generation is underdeveloped—could an autoregressive model achieve similar refinements?
2. **Quality:** The paper lacks detailed profiling of training and inference overhead; multi-round refinements increase model calls but latency and cost breakdowns are not provided.
3. **Significance:** It is unclear whether this search-driven scaling offers substantial advantage over traditional diffusion’s iterative re-noising and de-noising strategies.
4. **Clarity / Quality:** Ablation studies are limited; there is insufficient evidence isolating the contributions of the disruption operator, sampling steps, and post-processing to overall performance.
5. **Significance / Originality:** The approach, while effective for TSP and MIS, may require extensive engineering for other CO tasks, potentially limiting broader applicability.

---

> ### Author Rebuttal · Authors · 2025-07-30
>
> Thanks for the valuable comments, nice suggestions, and for acknowledging our work. Below we respond to your specific comments.
>
> ---
>
> ### ***160× Faster Than Gurobi with -0.06% Gap! Supplementary Experiments on a New Problem: Max Clique***
>
> At the beginning of the rebuttal, we would like to present an additional experiment evaluating GenSCO on the Max Clique problem to demonstrate its generality (as suggested by the reviewers), which we believe would be another firm empirical support of our methodology.  **Due to space limits, please refer to the beginning of the response to Reviewer 1QVA for the results.**
>
> **We are very surprised to find that GenSCO not only closes the gap to the powerful Gurobi solver but surpasses its solution quality with a -0.06% relative gap on the RB-Large dataset, while being over 160× faster computationally.** Compared to other learning-based solvers, this outcome marks a step-change, order-of-magnitude improvement. From the perspective of the gap metric, it has reduced the learning-based solvers' performance by an infinite-fold margin (from 0.90% to -0.06%).
>
> ---
>
> > **Q1: The motivation for using a rectified flow model for generation.**
>
> The motivation for using a lightweight rectified flow model is discussed in lines 135-149. Here we summarize as follows:
>
> **In essence, the motivation for using rectified flow is to create a more efficient, controllable, and scalable generative model that can be scaled more efficiently, allowing for maximizing the advanced scaling effects achieved by "generation as a search step".**
>
> Specifically, we are motivated by existing diffusion solvers' limitations:
> * Restricted Controllability: Relying on Gaussian noise initialization for *de novo* generation limits the ability to control intermediate states during generation.
> * Limited Exploitation of Prior Knowledge: It's difficult to effectively utilize existing solution knowledge.
> * Cost-Efficiency and Scalability: The generations follow complex, curved trajectories, which can be computationally expensive.
>
> GenSCO makes improvements accordingly, which we discuss in Q3.
>
> > **Q2: Profiling of training and inference overhead.**
>
> Thanks for the valuable question. The training costs of GenSCO on TSP-100/500/1000 are <= 2 days, <= 1 day, and <= 1 day on a GTX 4090 24G. For comparison, DIFUSCO takes 8.6 days, 2.7 days, and 5.1 days for training TSP-100/500/1000 on an A100 GPU, respectively.
>
> For inference cost, Fig. 2 shows the scaling curves of GenSCO, indicating the specific runtime and number of function calls of GenSCO for a given solving performance. Tables 1 and 4 show the runtimes for solving TSP and MIS with a varying number of search cycles. The inference cost of GenSCO is way smaller than that of previous state-of-the-art methods.
>
>
> > **Q3: Comparison to the traditional diffusion model’s iterative re-noising and de-noising process**
>
> In Figure 2(a), we illustrate the performance of the prior iterative re-noising and de-noising method (T2T) compared to simply scaling the de-noising step (DIFUSCO), as well as a comparison with our proposed GenSCO model. While T2T shows a clear performance improvement over DIFUSCO, this gain is limited by its methodology design based on single-round heavy generation. Our method aims to maximize the scaling effect of "generation as a search step", allowing us to integrate various search-friendly optimizations.
>
> The primary distinctions from previous approaches are as follows:
> 1. Focused Search Space: Our generation process is specifically designed to convert suboptimal solutions into optimal ones. This provides a more focused search range, and our bias-rectified training objective is specifically adapted for this goal.
> 2. Flow-Based Design for Speed: We employ a flow-based design, which ensures that generation follows a more direct trajectory, leading to faster inference.
> 3. Lightweight Transformer Architecture: We redesign the architecture using a lightweight Transformer. This maintains effectiveness while significantly improving efficiency.
> 4. Controllable Search Operator: During the inference phase, we design a controllable search operator that enables efficient scaling of the search process.
>
>
> > **Q4: Ablation studies are limited: the contributions of the disruption operator, sampling steps, and post-processing to overall performance.**
>
> Thanks for the nice advice. We supplement ablation studies as follows:
>
> 1. **w/  vs. w/o disruption operator**
>
> Using the disruption operator can effectively escape local optima and achieve continuous improvement. Without the disruption operator, we observe varying degrees of performance degradation, especially on TSP500, where the gains from increasing search cycles show significant stagnation.
>
> |Method|Disruption|TSP-500|TSP-1000|
> |-|-|-|-|
> |GenSCO (C=40)|w/o|16.551, 0.033%|23.137, 0.083%|
> || w/|16.549, 0.019%|23.133, 0.063%|
> |GenSCO (C=80)|w/o|16.551, 0.032%|23.134, 0.067%|
> || w/|16.548, 0.016%|23.131, 0.054%|
> |GenSCO (C=160)|w/o|16.551, 0.032%|23.131, 0.056%|
> ||w/|16.548, 0.012%|23.129, 0.046%|
>
> 2. **Effect of sampling steps**
>
> We observe that increasing num_sampling $t$ (at settings of $t=1, 2, 4, 6, 8, 16$) generally improved performance for num_cycles $C=40, 80, 160$, though the gains diminished in later stages. We ultimately chose $t=4$ and allocated more computational resources to scaling $C$.
>
> | C| t| Obj.| Gap| Time (s) |
> | - | -| -| -| -|
> | 40| 1   | 16.561  | 0.091% | 8.06|
> | | 2| 16.549  | 0.018% | 8.34 |
> | | 4| 16.548 | 0.013% | 18.38|
> | | 6| 16.548 | 0.013% | 26.88|
> | | 8| 16.548 | 0.012% | 35.60 |
> |80 | 1| 16.558  | 0.072% | 15.54|
> | | 2| 16.549  | 0.017% | 16.33|
> | | 4| 16.548  | 0.011% | 35.56|
> | | 6| 16.548  | 0.011% | 53.35|
> | |8| 16.547  | 0.010% | 70.66|
> |160 | 1| 16.556  | 0.059% | 31.47|
> | |2| 16.548  | 0.015% | 32.15|
> ||4| 16.548  | 0.010% | 70.62|
> | |6| 16.547  | 0.009% | 106.26|
> | |8| 16.547  | 0.009% | 140.92|
>
> 3. **Effect of post-processing**
>
> We use greedy decoding to guarantee the feasible output of the search operator, which is an indispensable part of this algorithm. Regarding 2-opt ablation for the TSP experiments, the ablation is already presented in Table 1.
>
> > **Q5: The approach, while effective for TSP and MIS, may require extensive engineering for other CO tasks, potentially limiting broader applicability.**
>
> The high-level algorithmic framework of our method shown in Fig. 1 is actually general, which can be applied to a wide range of COPs. To adapt methods to different problems, the major required effort is to customize specific components. In our framework, the primary problem-specific adaptations are the local search operators and the post-processing step. For many classical COPs, mature local search operators already exist. Even when they don't, creating simple operators to disrupt solutions is straightforward (as we do in MIS). For the post-processing step, it can be simply set as greedy decoding.
>
> To demonstrate the generality of our method, we conducted additional experiments on a new problem, the Maximum Clique Problem, within the limited rebuttal period. Please refer to the beginning of this rebuttal.
>
>
> > **Q6: Comparison of model parameter counts to ensure the comparisons are fair**
>
> Thanks for the advice on comparing the architectures. However, we must clarify that a direct comparison of parameter counts across all models in our study would be misleading due to fundamental differences in their underlying architectures. Unlike many NLP scenarios where comparisons are made between models of the same Transformer architecture, our baselines utilize diverse architectural designs. For instance, convolutional neural networks (GCNs) inherently possess fewer parameters than Transformer models thanks to repeated utilization of the same convolutional kernels. Directly comparing parameter counts between these distinct architectures would not accurately reflect their computational complexity or efficiency.
>
> Moreover, the lightweight Transformer architecture of our proposed Flow model represents a significant contribution. This emphasis on efficient and effective computation is crucial for maximizing the potential of our test-time scaling by "generation as a search step".
>
> To provide a more meaningful assessment of computational cost, we compare the computational time and performance of our proposed Transforme against a GCN model implemented under identical settings (consistent with baselines like DIFUSCO and T2T). **Due to space limits, please refer to the response to Reviewer 1QVA (Q3.3) for the results.** GCN requires >27 times computational cost for the same number of iterations, while yielding inferior results. Despite this, the GCN model's performance still surpassed that of previous baselines.
>
> > **Q7: Empirical evidence: solution-to-solution denoising vs. original noise-to-solution denoising**
>
> Thanks for the advice! We supplement the comparison to the SOTA noise-to-solution method, i.e., consistency model (Fast T2T), in our settings. **Due to space limits, please refer to the response to Reviewer 1QVA (Q3.1) for the results.** The results show the effectiveness of solution-to-solution denoising in our framework.
>
> > **Q8: In Lines 153–157, could the proposed disruption method generate invalid solutions?**
>
> The disruption operators we adopt are guaranteed to produce valid solutions. In the case of TSP, we use a 2-opt swap. For MIS, we randomly select a subset of nodes already in the independent set and remove them. Both of these operations are designed to perturb the solution within the feasible space.
>
> ---
>
> We hope this response provides the clarity you were seeking and fully addresses your concerns and that a more unified understanding and stronger consensus could be fostered around the value of our contribution to the community. Please do not hesitate to reach out if you have any additional questions or require further discussion. Thanks again for your valuable comments and suggestions.

---

> > ### Comment · Reviewer_vfnv · 2025-08-01
> > **Comment on Author Rebuttal**
> >
> > Thank you for the detailed response, which has effectively addressed my primary concerns. I will adjust my rating accordingly.

---

> > > ### Author Response · Authors · 2025-08-01
> > >
> > > Thank you sincerely for acknowledging our response. We greatly appreciate your thoughtful feedback and will ensure that the discussion and supplementary results are carefully incorporated into the final version of the paper. Once again, we are truly grateful for your time, valuable insights, and support throughout the review process.

---

### Official Review · Reviewer_sQhg · 2025-06-30

**Clarity:** 3
**Significance:** 3
**Originality:** 3
**Rating:** 5
**Confidence:** 4

**Summary:**

The authors present an improvement on the use of ML-based traveling salesman and maximally independent set solvers where they frame the use of a denoising diffusion model as a search operation to quickly explore the surrounding energy landscape and choose a good next update for the classical iterative energy-based solver. The model achieves this by using a Neural ODE formulation of the probability flow that is modeled with a simple transformer to predict the velocity field in the probability space. It is demonstrated on various tasks how this approach overcomes the denoising-step bottlenecks that many other diffusion-based solvers suffer. In particular, the model generalizes to some degree across scales (i.e. training on simpler problems generalizes to more complicated ones) and it seems to have some robustness in regard to the quality of the low-energy solution provided as a target.

**Questions:**

1.) How does the computational cost scale compared with just doing more solver steps in terms of FLOPS and energy. How big is the solution quality improvement?

2.) Regarding table 1: Why is GLOP and UDC not solved on TSP-100? Same for GenSCO in some cases? Can you elaborate on this?

3.) Did you ever try other neural network/ML models instead of the transformer to model the distribution?

**Ethical Concerns:**

["NO or VERY MINOR ethics concerns only"]

**Final Justification:**

All questions were properly addressed and the additional results further emphasize the relevance of this original work. My recommendation stands.

**Limitations:**

yes

**Paper Formatting Concerns:**

1.) Move related works section after introduction.

2.) Ln 115: Typo "he forward process"

3.) Ln 147: Grammar is completely off here. No third person singular "s" etc.

4.) Ln 152: Has to be Given "a" graph instance, ...

5.) Ln 186: "node number" should be replaced by "number of nodes" since that is what the authors are actually mean.

6.) Ln 296+297: Sentences just ends here. Something seems to be missing

7.) Ln 352: RB graphs and Erdos-Renyi graphs are introduced w/o explanation. How are they distinct from others?

**Quality:**

3

**Strengths And Weaknesses:**

**Strengths:**

1.) The work requires only a limited number of denoising steps (3-4) to already provide a measurable improvement on the performance of a solver. The linear scaling with denoising steps is impressive.

2.) The demonstration of the algorithm's ability to exceeding the plateau seems to be a major improvement over other data-driven/ML/DL-based solvers.

3.) The demonstration of cross-scale generalization of the method is another big plus of the paper.


**Weaknesses:**

1.) It seems that DIFUSCO is the biggest competitor to the presented algorithm. I think an in-depth comparison of the compute requirements in terms of FLOPS would have been appropriate. Similarly for a fair comparison to non-ML based solvers, FLOPS would have been a better and fairer metric since the ML-based solvers benefit from a high degree of parallelization that classical solvers, which typically run on CPUs can not take advantage of.

2.) Since the evaluations do not take too much time to run, it would have been nice to have 95% confidence intervals on measurement values since the performance of the solvers partially depend on the initialization of the starting point.

---

> ### Author Rebuttal · Authors · 2025-07-30
>
> Thanks for the valuable comments, nice suggestions, and for acknowledging our work. Below we respond to your specific comments.
>
>
> ---
>
> ### ***160× Faster Than Gurobi with -0.06% Gap! Supplementary Experiments on a New Problem: Max Clique***
>
> At the beginning of the rebuttal, we would like to present an additional experiment evaluating GenSCO on the Max Clique problem to demonstrate its generality (as suggested by the reviewers), which we believe would be another firm empirical support of our methodology.  **Due to space limits, please refer to the beginning of the response to Reviewer 1QVA for the results.**
>
> **We are very surprised to find that GenSCO not only closes the gap to the powerful Gurobi solver but surpasses its solution quality with a -0.06% relative gap on the RB-Large dataset, while being over 160× faster computationally.** Compared to other learning-based solvers, this outcome marks a step-change, order-of-magnitude improvement. From the perspective of the gap metric, it has reduced the learning-based solvers' performance by an infinite-fold margin (from 0.90% to -0.06%).
>
> ---
>
> > **Q1: FLOPs would have been a better and fairer metric since the ML-based solvers benefit from a high degree of parallelization that classical solvers.**
>
> Thank you for raising the point about a fair comparison metric. We appreciate your aim to find an equitable way to assess the computational complexity of traditional and learning-based solvers. However, we contend that a direct FLOPs (Floating Point Operations) comparison may omit one of the most significant advantages of learning-based solvers, i.e., highly parallelizable computation. Neural networks are inherently designed for highly parallelizable computation, and leveraging GPUs is a significant advantage of these methods. If we were to compare FLOPs, ML-based approaches would show an explosively high count due to extensive matrix operations. Yet, much of this computation completes rapidly because of its parallel nature, not serial execution.
>
> In response to your request, we compared the FLOPs of LKH and GenSCO on TSP100. Achieving the same 0.000% optimality gap, our perf tool measurements show LKH3 (max_trials=4096) with 357,716,687 FLOPs, while GenSCO (C=10) registered 1,624,669,544 FLOPs. Crucially, in terms of execution time, LKH took 14 minutes, whereas GenSCO completed the task in just 6 seconds.
>
> We believe this is a critical topic that we will discuss in the paper, hoping to foster more community discussion. We also look forward to seeing the field of operations research increasingly embrace the new vitality brought by GPU computing. Thanks again for this valuable point.
>
>
>
> > **Q2: It would have been nice to have confidence intervals on measurement values since the performance of the solvers partially depend on the initialization of the starting point.**
>
> Thanks for the suggestion! We agree that including confidence intervals is important. We have already run the GenSCO experiments 10 times to record the results and claculate the 95% confidence intervals of the GenSCO to show the performance variations of different initializations, where we discover consistent performance of GenSCO.
>
> Regarding the TSP-1000 testing times, our latest runs are slightly faster than what was presented in Table 1. This discrepancy likely stems from fluctuations in server computational resources or variations in CPU types and load (though the GPU type is consistent, i.e. RTX 4090) during this specific experiment set. We find that our newly measured results align with those obtained without 2-opt (where the computational cost of 2-opt is negligible), thus verifying the consistency of the current set of experimental results. We will update the original table accordingly to reflect these new, consistent results.
>
> | TSP-500 | Obj. | Gap (%) | Time (s) |
> | - | -| -| - |
> | C=10 | [16.549, 16.549] | [0.019, 0.020] | [4.901, 5.159]|
> | C=20 | [16.548, 16.549] | [0.015, 0.017] | [9.237, 9.275]|
> | C=40 | [16.548, 16.548] | [0.012, 0.014] | [17.966, 18.137] |
> | C=80| [16.548, 16.548] | [0.010, 0.012] | [35.523, 35.667] |
> | C=160| [16.547, 16.547] | [0.009, 0.010] | [70.705, 70.876] |
>
> | TSP-1000 | Obj. | Gap (%) | Time (s)  |
> | - | -| -| -|
> | C=10| [23.133, 23.134] | [0.063, 0.067] | [12.562, 12.691]|
> | C=20| [23.131, 23.131] | [0.054, 0.056] | [24.155, 24.432]|
> | C=40| [23.129, 23.129] | [0.046, 0.047] | [47.435, 47.810]|
> | C=80| [23.127, 23.127] | [0.039, 0.040] | [94.012, 94.349]|
> | C=160| [23.126, 23.126] | [0.034, 0.035] | [187.295, 187.693] |
>
> > **Q3: How does the computational cost scale compared with just doing more solver steps in terms of FLOPS and energy. How big is the solution quality improvement?**
>
> Thank you for this insightful question. As shown in Fig. 2(b), scaling search cycles with GenSCO leads to a more stable performance and a higher quality of solution. This approach effectively overcomes the performance plateau that occurs when you simply increase the number of inference steps in a single generation round.
>
> To compare the computational cost of these two methods (scaling denoising steps vs. scaling search cycles), we use the number of function evaluations (NFE) as the primary metric, which represents the number of times the neural network is executed. This is the unit used for the x-axis in Fig. 2(b). When measured by NFE, both methods have an identical computational cost in terms of FLOPS and energy consumption on the GPU. The main operations for both are the parallelizable matrix multiplications of the neural network, which are handled by the GPU. Any additional operations on the CPU are computationally negligible in a Big O notation sense. Therefore, from a computational complexity standpoint, the cost scales identically for both increasing search cycles and increasing solver steps.
>
>
> > **Q4: Regarding table 1: Why is GLOP and UDC not solved on TSP-100? Same for GenSCO in some cases?**
>
> Both GLOP and UDC are divide-and-conquer methods designed specifically for very large-scale problems. Their original papers did not include implementations or results for the smaller TSP-100 scale, which is why those cells are empty.
>
> The blank cells for GenSCO on TSP-100 are because GenSCO had already found the optimal solution at 10 cycles. Since increasing the number of cycles beyond this point would not have improved the solution quality, we omitted the redundant data for clarity.
>
> > **Q5: Did you ever try other neural network/ML models instead of the transformer to model the distribution?**
>
> Thanks for the nice advice. We supplement ablation studies as follows:
>
> 1. **Flow model vs. consistency model (Fast T2T [1])**
>
> We compare the flow model and the consistency model on TSP-500 under the same implementation and number of forward function evaluations (NFEs). We treat the consistency model as an alternative diffusion model variant and enhance its competitiveness by applying more 2-opt cycles, though at the cost of increased runtime. The results demonstrate the effectiveness of our proposed flow model over the previous consistency model design.
>
> | NFEs | Method | Obj. | Gap | Time |
> | :--- | :--- | :--- | :--- | :--- |
> | 40 | Consistency | 16.598 | 0.313% | 7s |
> |  | Consistency 2opt | 16.550 | 0.028% | 23s |
> |  | Flow | 16.550 | 0.023% | 5s |
> |  | Flow 2opt | 16.549 | 0.019% | 5s |
> | 80 | Consistency | 16.590 | 0.265% | 14s |
> |  | Consistency 2opt | 16.549 | 0.022% | 46s |
> |  | Flow | 16.549 | 0.020% | 9s |
> |  | Flow 2opt | 16.548 | 0.016% | 9s |
> | 160 | Consistency | 16.584 | 0.233% | 28s |
> |  | Consistency 2opt | 16.549 | 0.021% | 1m33s |
> |  | Flow | 16.548 | 0.016% | 18s |
> |  | Flow 2opt | 16.548 | 0.012% | 18s |
>
>
> 3. **Architecture: Transformer vs. GCN**
>
> We implement the GCN following the specific settings in DIFUSCO [2] and Fast T2T [1] as an alternative backbone for the flow model. The GCN implementation requires significantly more runtime for the same iterations, whereas the proposed Transformer architecture's efficiency enables the maximization of scaling benefits.
>
>
> |Method| Backbone| TSP-500| TSP-1000|
> |-|-|-|-|
> |GenSCO (C=40)  | GCN| 16.556, 0.060%, 1m45s | 23.157, 0.167%, 6m50s  |
> || transformer | 16.549, 0.019%, 5s| 23.133, 0.063%, 16s|
> | GenSCO (C=80)  | GCN| 16.553, 0.045%, 3m25s | 23.147, 0.127%, 13m40s |
> || transformer | 16.548, 0.016%, 9s| 23.131, 0.054%, 30s|
> | GenSCO (C=160) | GCN| 16.552, 0.037%, 6m50s | 23.140, 0.097%, 27m20s |
> || transformer | 16.548, 0.012%, 18s| 23.129, 0.046%, 58s|
>
>
> > **Q6: Ln 352: RB graphs and Erdos-Renyi graphs are introduced w/o explanation. How are they distinct from others?**
>
> Thank you for this valuable question. We will add an explanation to the paper to clarify the distinction.
>
> RB graphs are a class of random graphs specifically designed to generate challenging problem instances stemming from constraint satisfaction problems. The RB model creates instances that are known to be computationally difficult to solve, making them ideal for evaluating the robustness of algorithms.
>
> Erdos-Renyi (ER) graphs, on the other hand, are a classical and foundational model of random graphs. In this model, edges are formed independently and with a uniform probability between all pairs of nodes. They serve as a standard baseline but are not specifically tuned to produce hard problem instances like RB graphs.
>
>
> > **Q7: Other paper formatting concerns.**
>
> Thanks for the detailed review. We will update our script according to your valuable comments.
>
> ---
>
> We sincerely hope this response clarifies your concerns. We also hope a unified understanding and a stronger consensus can be fostered among the reviewers regarding the value of our contribution to the community. Thanks again for your valuable review and support.
>
> ---
>
> [1] Fast T2T: Optimization Consistency Speeds Up Diffusion-Based Training-to-Testing Solving for Combinatorial Optimization. NeurIPS 2024.
>
> [2] DIFUSCO: Graph-based Diffusion Solvers for Combinatorial Optimization. NeurIPS 2023.

---

> ### Comment · Reviewer_sQhg · 2025-08-02
> **Reply to the Rebuttal**
>
> I thank the authors for this highly detailed response! All my questions were answered to my satisfaction. I further thank the authors for addressing the FLOPs as a fairer metric and reporting them for a specific problem. It seems that the difference is about a factor of 4x which is acceptable for an orders-of-magnitude speedup. I believe this work is a valuable contribution and I maintain my score.
>
> I strongly advise the authors to use the additional results generated for the rebuttals in the main body and appendix of the paper. They significantly contribute to a better and more well-rounded understanding of the method.

---

> > ### Author Response · Authors · 2025-08-03
> >
> > Thank you for your positive feedback on our response and for its acknowledgment. Your recognition of our work as a valuable contribution is supportive and encouraging. We fully agree that the additional results significantly enhance our work and will ensure they are carefully integrated into the final version.
> >
> > Our sincere thanks again for your valuable time, insightful comments, and ongoing support during this review process.

---

### Official Review · Reviewer_5uKj · 2025-07-02

**Clarity:** 3
**Significance:** 3
**Originality:** 3
**Rating:** 5
**Confidence:** 4

**Summary:**

This paper introduces GenSCO, a diffusion-based solver that treats generation as a search operator within the near-optimal solution space. Extensive experiments on classic TSP and MIS problems demonstrate the superiority of GenSCO.

**Questions:**

- Line 302 states that 8 GenSCO operators are run in parallel. I would like to know whether the runtimes reported in Table 1 and Table 4 reflect this parallel execution. If so, the comparison with other solvers may not be entirely fair.
- Can authors provide more discussions on which other more complex combinatorial optimization problem GenSCO can solve?
- GenSCO is a test-time scaling method. Why don’t you compare it with using Euclidean distance as prior information instead of a diffusion model, as done in [1]? I’m unsure about the effectiveness of the learning component, especially since neural methods are a primary focus of this conference.
[1] Rethinking the "Heatmap + Monte Carlo Tree Search'' Paradigm for Solving Large Scale TSP

**Ethical Concerns:**

["NO or VERY MINOR ethics concerns only"]

**Final Justification:**

The additional materials have addressed my concerns and so I raised my score accordingly.

**Limitations:**

Main limitations lie at the limited datasets for the comparison with the exiting works

**Quality:**

3

**Strengths And Weaknesses:**

Strengths:
- Novel Approach: The paper introduces a novel approach called GenSCO, which systematically exploits the learned distribution of high-quality solutions.
- Computational results: The paper claims to GenSCO achieve significant speedup on TSP and MIS.

Weaknesses:
- Limited Benchmark Problems: While the authors present results on TSP and MIS benchmarks, it would be helpful to evaluate GenSCO on more complex combinatorial optimization problems to better demonstrate its generalizability.
- There is a misspelled word in line 138.

---

> ### Author Rebuttal · Authors · 2025-07-29
>
> Thanks for the valuable comments, nice suggestions, and for acknowledging our work. Below we respond to your specific comments.
>
> ---
>
> ### ***160× Faster Than Gurobi with -0.06% Gap! Supplementary Experiments on a New Problem: Max Clique***
>
> At the beginning of the rebuttal, we would like to present an additional experiment evaluating GenSCO on the Max Clique problem to demonstrate its generality (as suggested by the reviewers), which we believe would be another firm empirical support of our methodology. **We are indeed very surprised to find that our method outperforms existing state-of-the-art learning-based and classic solvers on this problem by a large margin, and it did so with little hyper-parameter tuning.** We believe this strongly supports the versatility of our approach. The results are as follows:
>
> |                          |           | RB-Small  |           |           | RB-Large  |           |
> | ------------------------ | --------- | --------- | --------- | --------- | --------- | --------- |
> | **Method**              | **Obj.↑** | **Drop↓** | **Time↓** | **Obj.↑** | **Drop↓** | **Time↓** |
> | Gurobi                   | **19.08***    | **0.00%**     | **7.5m**      | **40.18***    | **0.00%**     | **38.4h**     |
> | Meta-EGN [1]              | 17.51     | 8.30%     | 2.3m      | 33.79     | 15.49%    | 4.5m      |
> | DiffUCO [2]               | 16.21     | 12.53%    | 11.8m     | --        | --        | --        |
> | COExpander [3] more steps | 19.00     | 0.50%     | 5.4m      | 39.06     | 2.99%     | 53.0m     |
> | COExpander [3] bs         | 18.98     | 0.68%     | 30s       | 39.88     | 0.90%     | 3.1m      |
> | GenSCO (C=500)           | 19.08     | 0.03%     | 1.1m      | 40.18     | 0.01%     | 6.9m      |
> | GenSCO (C=1k)            | **19.08**     | **0.00%**     | **2.2m**      | **40.20**     | **-0.06%**    | **13.8m**    |
>
> For the Maximum Clique problem, we compared our method against baselines using the best parameters reported in their original papers. For Gurobi, we imposed a time limit of 300 seconds per instance. All methods are evaluated over a total of 500 instances and other experimental settings exactly follow COExpander [3].
>
> **We are very surprised to find that GenSCO not only closes the gap to the powerful Gurobi solver but surpasses its solution quality with a -0.06% relative gap on the RB-Large dataset, while being over 160× faster computationally.** Compared to other learning-based solvers, this outcome marks a step-change, order-of-magnitude improvement. From the perspective of the gap metric, it has reduced the learning-based solvers' performance by an infinite-fold margin (from 0.90% to -0.06%).
>
> ---
>
> > **Q1: Can authors provide more discussions on which other more complex combinatorial optimization problems GenSCO can solve?**
>
> Similar to previous generative model solvers, our framework uses a "heatmap + post-processing" approach to solve problems. This framework can, in principle, be applied to any problem that meets the following two conditions:
>
> 1. The decision variables are limited to a countable and finite set of values. This can cover edge-selecting problems like TSP, and node-selecting problems like MIS, Max Clique.
> 2. A problem-specific post-processing procedure is required to transform the prediction into feasible solutions. This prerequisite can be easily met with the greedy decoding strategy, which sequentially inserts variables (edges or nodes) with the highest confidence if there are no conflicts.
>
> In theory, any problem satisfying these conditions can be tackled by GenSCO. However, a limitation arises with problems that have highly complex constraints, such as CVRP. Our current approach relies on a combination of a "soft" constraint satisfaction learned by the model and a "hard" guarantee from the post-processing step. Since the model's prediction process doesn't explicitly incorporate constraints, the prediction accuracy can degrade when dealing with intricate dependencies. Improving our method to handle such complex constraints more effectively is a key direction for future work.
>
>
> To demonstrate the generality of our method, we conducted additional experiments on a new problem, the Maximum Clique Problem, within the limited rebuttal period. Please refer to the results in the beginning of the rebuttal.
>
>
>
> > **Q2: Line 302 states that 8 GenSCO operators are run in parallel. I would like to know whether the runtimes reported in Table 1 and Table 4 reflect this parallel execution. If so, the comparison with other solvers may not be entirely fair.**
>
>
> The runtimes reported in Tables 1 and 4 do reflect the parallel execution of 8 GenSCO operators. However, we note that the core logic of our methodology is: 1) to explore a new solving paradigm (generation as a search operator) that offers better scaling effects, and 2) to maximize the use of the proposed operator to significantly boost solving performance. In this design, the neural network's role is lightweight, with a low number of inference steps per generation, which is designed to reallocate the computation to more search cycles. Compared to previous methods that rely on the accuracy of single-round heavy generation, GenSCO's single neural network inference has a low GPU computational density. Parallelism is necessary to ensure GenSCO fully utilizes the available GPU power, making the computational cost comparable to other methods that rely on "heavy" networks for a single model inference.
>
> To ensure full clarity, we will add a statement to the paper explaining this parallel execution. Here, we also include results for 1, 2, and 4 parallel operators to show the effect of varying parallel runs. As seen, even with the num_runs=1 setting, GenSCO still significantly outperforms other baselines (without full GPU utilization).
>
> | Cycles | runs | TSP-500 Obj. | TSP-500 Gap | TSP-500 Time(s) | TSP-1000 Obj. | TSP-1000 Gap | TSP-1000 Time(s) |
> | :--- | :--- | :--- | :--- | :--- | :--- | :--- | :--- |
> | **C = 40** | 1 | 16.550 | 0.028 | 2.408 | 23.137 | 0.083 | 7.609 |
> | | 2 | 16.549 | 0.021 | 4.383 | 23.133 | 0.065 | 12.372 |
> | | 4 | 16.548 | 0.016 | 9.248 | 23.130 | 0.053 | 24.164 |
> | | 8 | 16.548 | 0.013 | 17.974 | 23.129 | 0.047 | 47.468 |
> | **C = 80** | 1 | 16.549 | 0.021 | 4.365 | 23.134 | 0.067 | 14.012 |
> | | 2 | 16.549 | 0.016 | 8.333 | 23.131 | 0.055 | 23.834 |
> | | 4 | 16.548 | 0.012 | 17.904 | 23.129 | 0.047 | 47.327 |
> | | 8 | 16.548 | 0.011 | 35.541 | 23.127 | 0.040 | 93.989 |
> | **C = 160** | 1 | 16.548 | 0.016 | 8.226 | 23.131 | 0.056 | 27.741 |
> | | 2 | 16.548 | 0.013 | 16.160 | 23.129 | 0.047 | 46.770 |
> | | 4 | 16.548 | 0.011 | 35.375 | 23.128 | 0.041 | 93.904 |
> | | 8 | 16.548 | 0.010 | 70.923 | 23.126 | 0.034 | 187.634 |
>
>
>
> > **Q3: Why don’t you compare it with using Euclidean distance as prior information instead of a diffusion model, as done in [4]? I’m unsure about the effectiveness of the learning component, especially since neural methods are a primary focus of this conference.**
>
>
> Thanks for the valuable question. First and foremost, we would like to clarify a key distinction. Paper [4] discusses the role of neural networks in augmenting strong search methods like MCTS. MCTS is a powerful technique, and introducing neural network predictions as a guidance mechanism doesn't always guarantee a significant, or even limited, improvement. In contrast, GenSCO is an end-to-end neural solver, aligning with established works like DIFUSCO [5], T2T [6], AM [7], and POMO [8]. These methods rely solely on learned representations, using only lightweight post-processing (e.g., greedy decoding or optional 2-opt for TSP).
>
> The solving capability of GenSCO stems entirely from its learned neural representations. Unlike [4], there is no search framework (e.g., MCTS) to refine weak priors. Replacing the diffusion model with Euclidean distances would reduce the approach to nearest insertion (a naive baseline), leading to >20% performance gaps on TSP100, which would be unacceptable.
>
> ---
>
> We hope this response provides the clarity you were seeking and fully addresses your concerns. We would be truly grateful if, with further consideration, a more unified understanding and stronger consensus regarding the value of our contribution to the community could be fostered. We welcome any further questions and are available for additional discussion. Thanks again for your valuable comments and suggestions.
>
> ---
>
> [1] Unsupervised Learning for Combinatorial Optimization Needs Meta Learning. ICLR 2023.
>
> [2] A Diffusion Model Framework for Unsupervised Neural Combinatorial Optimization. ICML 2024.
>
> [3] COExpander: Adaptive Solution Expansion for Combinatorial Optimization. ICML 2025.
>
> [4] Rethinking the "Heatmap + Monte Carlo Tree Search" Paradigm for Solving Large Scale TSP.
>
> [5] DIFUSCO: Graph-based Diffusion Solvers for Combinatorial Optimization. NeurIPS 2023.
>
> [6] T2T: From Distribution Learning in Training to Gradient Search in Testing for Combinatorial Optimization. NeurIPS 2023.
>
> [7] Attention, Learn to Solve Routing Problems! ICLR 2019.
>
> [8] POMO: Policy Optimization with Multiple Optima for Reinforcement Learning. NeurIPS 2020.

---

> > ### Author Response · Authors · 2025-08-08
> > **A Kind Reminder of Mandatory Acknowledgement**
> >
> > Dear Reviewer 5uKj,
> >
> > Thank you once again for your thorough review and insightful comments on our work. As the discussion period draws to a close, we want to kindly follow up to confirm whether you’ve had the opportunity to revisit our detailed rebuttal and supplementary materials. We sincerely appreciate your feedback and would be grateful to hear whether our responses have adequately addressed your concerns or if there are additional points you’d like us to clarify.
> >
> > We sincerely hope that our work will breathe new vitality into these essential and well-established optimization fields. Through the community’s collective efforts, we also aspire to spark a potential paradigm shift in problem-solving amid the AI era. We commit to making our source code publicly available upon acceptance.
> >
> > Our sincere thanks again for your valuable time, constructive suggestions, and ongoing support throughout this review process.
> >
> > Best regards,
> >
> > The Authors

---

> > ### Comment · Reviewer_5uKj · 2025-08-09
> >
> > Thanks a lot for the detailed response. The additional materials have improved the presentation of the paper and solidify the contribution of  this work. The proposed search operator has its power to improve the solver of the CPs. I will adjust my score accordingly and suggest the author include the new materials into their follow-up version of the submission.

---

### Official Review · Reviewer_1QVA · 2025-07-03

**Clarity:** 2
**Significance:** 3
**Originality:** 3
**Rating:** 5
**Confidence:** 2

**Summary:**

This paper introduces GenSCO, the key idea is to start from suboptimal solutions (not random noise) and train the model to recover the optimal solution. This change allows the model to explore a smaller, more relevant part of the solution space, leading to better performance. Experiments on TSP and MIS problems show that GenSCO outperforms strong baselines in both quality and speed.

**Questions:**

- Since the new method requires meaningful disruption of existing solutions and the way to achieve this is problem dependent, how could the proposed method be possibly applied to broader CO problems beyond TSP and MIS to improve its generality?
- Is there any ablation study to analyze how much each part of the system (e.g., flow model, search cycles, architecture) contributes to the final result?
- In Section 3.2, during each search cycle, the disruption rate for MIS seems to be an arbitrary constant from 25% to 40%. Is there any reasoning or justification on how the disruption rate is set?

**Ethical Concerns:**

["NO or VERY MINOR ethics concerns only"]

**Final Justification:**

The authors have thoroughly addressed the major concerns raised in the review. The paper is solid with additional experiment and ablation study

**Limitations:**

yes

**Quality:**

3

**Strengths And Weaknesses:**

Strengths:
- New training setup: training the model to go from suboptimal to optimal solutions (instead of noise to solution) is a clever shift that makes the search more efficient.
- Strong results: GenSCO beats both neural baselines and classic solvers like LKH3 in quality and runtime on multiple problem sizes.

Weaknesses
- Table formatting and precision: some tables like Table 1 show the same objective value but different optimality gaps, which is confusing.
- Limited problem coverage: the experiments focus only on TSP and MIS. These are standard, but don’t show how the method works on other types of problems.
- No component ablation: it’s unclear how much each part of the system (e.g., flow model, search cycles, architecture) contributes to the final result.

---

> ### Author Rebuttal · Authors · 2025-07-29
>
> Thanks for the valuable comments, nice suggestions, and for acknowledging our work. Below we respond to your comments.
>
> ---
>
> ### ***160× Faster Than Gurobi with -0.06% Gap! Supplementary Experiments on a New Problem: Max Clique***
>
> At the beginning of the rebuttal, we would like to present an additional experiment evaluating GenSCO on the Max Clique problem to demonstrate its generality (as suggested by the reviewers), which we believe would be another firm empirical support of our methodology. **We are indeed very surprised to find that our method outperforms existing state-of-the-art learning-based and classic solvers on this problem by a large margin, and it did so with little hyper-parameter tuning.** We believe this strongly supports the versatility of our approach. The results are as follows:
>
> |   | | RB-Small  || | RB-Large  | |
> | - | - | - | - | - | -| -|
> | **Method**  | **Obj.↑** | **Drop↓** | **Time↓** | **Obj.↑** | **Drop↓** | **Time↓** |
> | Gurobi  | **19.08***    | **0.00%**     | **7.5m**      | **40.18***    | **0.00%**     | **38.4h**     |
> | Meta-EGN [1]  | 17.51     | 8.30%     | 2.3m      | 33.79     | 15.49%    | 4.5m      |
> | DiffUCO [2] | 16.21     | 12.53%    | 11.8m     | --        | --        | --        |
> | COExpander [3] more steps | 19.00     | 0.50%     | 5.4m      | 39.06     | 2.99%     | 53.0m     |
> | COExpander [3] bs | 18.98     | 0.68%     | 30s       | 39.88     | 0.90%     | 3.1m      |
> | GenSCO (C=500) | 19.08     | 0.03%     | 1.1m      | 40.18     | 0.01%     | 6.9m      |
> | GenSCO (C=1k) | **19.08**     | **0.00%**     | **2.2m**      | **40.20**     | **-0.06%**    | **13.8m**    |
>
> For the Maximum Clique problem, we compared our method against baselines using the best parameters reported in their original papers. For Gurobi, we imposed a time limit of 300 seconds per instance. All methods are evaluated over a total of 500 instances and other experimental settings exactly follow COExpander [3].
>
> **We are very surprised to find that GenSCO not only closes the gap to the powerful Gurobi solver but surpasses its solution quality with a -0.06% relative gap on the RB-Large dataset, while being over 160× faster computationally.** Compared to other learning-based solvers, this outcome marks a step-change, order-of-magnitude improvement. From the perspective of the gap metric, it has reduced the learning-based solvers' performance by an infinite-fold margin (from 0.90% to -0.06%).
>
> ---
>
> > **Q1: Table formatting and precision: Table 1 shows the same objective value but different optimality gaps.**
>
> The value discrepancy in Table 1 arises from numerical precision. The objective values in the table are rounded to two decimal places for presentation, whereas the optimality gaps are calculated using the full-precision values.
>
> For instance, consider the TSP-100 dataset, where the optimal solution is 7.75585. A solution with a length of 7.756 and another with 7.764 would both be displayed as "7.76" after rounding. However, their true difference is accurately captured by the optimality gap, which is calculated with the unrounded values, resulting in gaps of 0.002% and 0.105%, respectively.
>
>
> > **Q2: Limited problem coverage: the experiments focus only on TSP and MIS. These are standard, but don’t show how the method works on other types of problems.**
>
> Thanks for the advice. To demonstrate the generality of our method, we conducted additional experiments on a new problem, the Maximum Clique Problem, within the limited rebuttal period. Please refer to the full results at the beginning of the rebuttal.
>
>
> > **Q3: Is there any ablation study to analyze how much each part of the system (e.g., flow model, search cycles, architecture) contributes to the final result?**
>
> Thanks for the nice advice. We supplement ablation studies as follows, and we will include all these results in our paper.
>
> 1. **Flow model vs. consistency model (Fast T2T [4])**
>
> We compare the flow model and the consistency model on TSP-500 under the same implementation and number of forward function evaluations (NFEs). We treat the consistency model as an alternative diffusion model variant and enhance its competitiveness by applying more 2-opt cycles, though at the cost of increased runtime. The results demonstrate the effectiveness of our proposed flow model over the previous consistency model design.
>
>
> | NFEs | Method | Obj. | Gap | Time |
> | :--- | :--- | :--- | :--- | :--- |
> | 40 | Consistency | 16.598 | 0.313% | 7s |
> |  | Consistency 2opt | 16.550 | 0.028% | 23s |
> |  | Flow | 16.550 | 0.023% | 5s |
> |  | Flow 2opt | 16.549 | 0.019% | 5s |
> | 80 | Consistency | 16.590 | 0.265% | 14s |
> |  | Consistency 2opt | 16.549 | 0.022% | 46s |
> |  | Flow | 16.549 | 0.020% | 9s |
> |  | Flow 2opt | 16.548 | 0.016% | 9s |
> | 160 | Consistency | 16.584 | 0.233% | 28s |
> |  | Consistency 2opt | 16.549 | 0.021% | 1m33s |
> |  | Flow | 16.548 | 0.016% | 18s |
> |  | Flow 2opt | 16.548 | 0.012% | 18s |
>
>
> 2. **Effect of Search Cycles**
>
> Please refer to GenSCO results in Table 1 and Table 4, where we report the solving performance and runtimes of GenSCO with varying search cycles. As shown, scaling search cycles results in stable improvements.
>
>
> 3. **Architecture: Transformer vs. GCN**
>
> We implement the GCN following the specific settings in DIFUSCO [5] and Fast T2T [4] as an alternative backbone for the flow model. The GCN implementation takes much more runtime for the same iterations, and the efficiency of the proposed Transformer architecture allows for the maximization of the scaling benefits.
>
>
> | Method| Backbone| TSP-500| TSP-1000|
> | ----- | -- | --- | --- |
> | GenSCO (C=40)  | GCN| 16.556, 0.060%, 1m45s | 23.157, 0.167%, 6m50s  |
> || transformer | 16.549, 0.019%, 5s    | 23.133, 0.063%, 16s    |
> | GenSCO (C=80)  | GCN| 16.553, 0.045%, 3m25s | 23.147, 0.127%, 13m40s |
> || transformer | 16.548, 0.016%, 9s    | 23.131, 0.054%, 30s    |
> | GenSCO (C=160) | GCN| 16.552, 0.037%, 6m50s | 23.140, 0.097%, 27m20s |
> || transformer | 16.548, 0.012%, 18s   | 23.129, 0.046%, 58s    |
>
>
>
> > **Q4: Since the new method requires meaningful disruption of existing solutions in a problem-dependent way, how could the proposed method be possibly applied to broader CO problems beyond TSP and MIS to improve its generality?**
>
> Our method’s generality lies in its high-level algorithmic framework, as shown in Fig. 1, which can be applied to a wide range of COPs. To adapt methods to different problems, it is necessary and actually standard to customize specific components. In our framework, the primary problem-specific adaptations are the local search operators and the post-processing step. Importantly, these customizations don’t affect the core neural network training method or the overall framework's operation. For many classical COPs, mature local search operators already exist. Even when they don't, creating simple operators to disrupt solutions is straightforward (as we do in MIS). For the post-processing step, it can be simply set as greedy decoding.
>
> This need for problem-specific adaptation is standard for various methods (you have to make adaptations for different problems for different inputs/constraints/characteristics). For a more closely related diffusion-based example, T2T [6] must construct a gradient source loss for different problems by converting the original objective and constraints. Similarly, it also needs to consider different architectures and post-processing methods. This also holds for methods like DIMES [7], DIFUSCO [5], Fast T2T [4] (architectures and post-processing), AM [8], POMO [9] (input processing and intermediate action space control).  Despite these required customizations, these ML algorithms are still considered general at the architectural level, and our method is no different.
>
> Please see Q2 for the supplementary experiments on a new problem, Max Clique, which achieves advantages over existing state-of-the-art learning-based and classic solvers by a large margin.
>
> > **Q5: In Section 3.2, during each search cycle, the disruption rate for MIS seems to be an arbitrary constant from 25% to 40%. Is there any reasoning or justification on how the disruption rate is set?**
>
> The disruption rate is a hyperparameter that controls the size of the local search neighborhood, which is crucial for maintaining search efficiency. The range of 25% to 40% was determined experimentally to be a relatively more effective range for MIS. An intuition for selecting this parameter is that we initially refer to the 25% - 75% 2opt disruption in TSP. Considering that flipping variables in MIS has a greater impact on the solution, a smaller upper limit for the disruption rate is chosen.
>
> ---
>
> We hope this response clarifies your understanding and fully addresses your concerns and that a more unified understanding and stronger consensus could be fostered regarding the value of our contribution to the community. Please do not hesitate to reach out with any additional questions or for further discussion. Thanks again for your valuable comments and suggestions.
>
> ---
>
> [1] Unsupervised Learning for Combinatorial Optimization Needs Meta Learning. ICLR 2023.
>
> [2] A Diffusion Model Framework for Unsupervised Neural Combinatorial Optimization. ICML 2024.
>
> [3] COExpander: Adaptive Solution Expansion for Combinatorial Optimization. ICML 2025.
>
> [4] Fast T2T: Optimization Consistency Speeds Up Diffusion-Based Training-to-Testing Solving for Combinatorial Optimization. NeurIPS 2024.
>
> [5] DIFUSCO: Graph-based Diffusion Solvers for Combinatorial Optimization. NeurIPS 2023.
>
> [6] T2T: From Distribution Learning in Training to Gradient Search in Testing for Combinatorial Optimization. NeurIPS 2023.
>
> [7] DIMES: A Differentiable Meta Solver for Combinatorial Optimization Problems. NeurIPS 2022.
>
> [8] Attention, Learn to Solve Routing Problems! ICLR 2019.
>
> [9] POMO: Policy Optimization with Multiple Optima for Reinforcement Learning. NeurIPS 2020.

---

> > ### Comment · Reviewer_1QVA · 2025-08-05
> >
> > Thanks for the detailed response, especially the extra experiment on the extra problem and breakdown on each component. They have effectively addressed my questions, and I will adjust my rating accordingly. I highly suggest the authors to include the additional results into the next version of the paper

---

> > > ### Author Response · Authors · 2025-08-06
> > >
> > > Thank you for your positive feedback on our response and for raising the score! We fully agree that the additional results significantly enhance our work and will ensure they are carefully integrated into the final version.
> > >
> > > It is our sincere hope that our work will inject new vitality into the ML for Optimization community and further demonstrate the value of machine learning for traditional optimization fields. We also hope that through the community's collective efforts, we can spark a potential paradigm evolution in problem-solving amid the wave of the AI era.
> > >
> > > Our sincere thanks again for your valuable time, constructive suggestions, and ongoing support throughout this review process.

---

### Note · Authors · 2025-08-12

We sincerely thank the chairs and reviewers for their time, feedback, and constructive suggestions.

In the **initial review (scores: 4, 4, 4, 5)**, reviewers acknowledged our work's novelty (1QVA, 5uKj, sQhg, vfnv), significance (1QVA, sQhg, vfnv), and quality (1QVA, 5uKj, sQhg, vfnv). They praised our method as a "clever shift in training designs" (1QVA) that "systematically exploits the learned distribution of high-quality solutions" (5uKj). Our empirical results were recognized as "strong," delivering "significant," "major," and "impressive" improvements over both neural baselines and classic solvers like LKH3 (1QVA, 5uKj, sQhg). Key technical contributions, such as the "impressive" linear scaling (sQhg), the "cost-efficient" pipeline redesign (vfnv), and "strong cross-scale generalization" (sQhg), were highlighted as major strengths.

---

During the discussion period, we focused on addressing the primary concerns:

1. **Generalizability to Other Problems:** We clarified GenSCO’s scope, supplementing new results on Max Clique, where it outperformed existing state-of-the-art learning-based and classic solvers by a large margin (160× faster than Gurobi with -0.06% Gap).

2. **Missing Some Ablation Studies:** We now include comprehensive ablation studies quantifying the impact of key components like the rectified flow model, network architecture, disruption operator, and the number of sampling steps, showing the performance gains from our design choices.

3. **Clarification of Details:** We’ve added further details, analyses, and experiments to address specific questions, including profiling of training and inference overhead, rationale for our generative model choice, computational costs, confidence intervals, and justification for various designs, etc.

---

**We are grateful that all reviewers acknowledged our response, and that the three reviewers who had initially assigned a score of '4' expressed intention to raise their ratings.** They agreed that the new results and discussions significantly enhance the paper, and we will ensure they will be fully integrated.

We hope that this work can introduce new vitality to the ML4Opt community. We believe that through the collective efforts of the community, ML can continue to unlock new frontiers in optimization and help spark a paradigm shift in problem-solving amid the wave of the AI era.

Our sincere thanks again to the chairs and reviewers! Your feedback has strengthened this paper significantly.

---

### Decision · Program_Chairs · 2025-09-17

**Decision:**

Accept (poster)

**Comment:**

This paper introduces GenSCO, a novel framework for scaling diffusion-based solvers in combinatorial optimization by treating generation as a search operator rather than a complete solving process. The method integrates local search disruption, diffusion sampling, and a rectified flow model with a lightweight transformer, enabling efficient exploration of the solution space. Strengths include its clear conceptual novelty, rigorous training design, and compelling experimental results demonstrating orders-of-magnitude improvements over prior neural methods and even substantial speedups against the strong classical solver LKH3. Weaknesses noted by reviewers focused on clarity of exposition and some missing ablations, but these were largely addressed during rebuttal with clarifications and additional evidence. Reviewers have actively participated in the discussion and reached a clear concensus to accept this paper. Overall, I recommend acceptance.